

# Weaving of biomineralization framework in rotaliid foraminifera: Implications for paleoenvironmental reconstructions

Yukiko Nagai[1,2*], Katsuyuki Uematsu[3], Chong Chen[2], Ryoji Wani[4], Jarosław Tyszka[5] and Takashi Toyofuku[2,6]

5  [1]Graduate School of Environment and Information Sciences, Yokohama National University, 79-7, Tokiwadai, Hodogaya-ku, Yokohama, 240-8501, Japan
[2]Japan Agency for Marine-Earth Science and Technology (JAMSTEC), Natsushima-cho 2-15, Yokosuka, 237-0061, Japan
[3]Marine Works Japan Ltd., 3-54-1 Oppama-higashi, Yokosuka 237-0063, Japan
[4]Faculty of Environment and Information Sciences, Yokohama National University, 79-7, Tokiwadai, Hodogaya-ku, 10  Yokohama, 240-8501, Japan
[5]ING PAN - Institute of Geological Sciences, Polish Academy of Sciences, Research Centre in Cracow, Senacka 1, 31-002 Kraków, Poland
[6]Tokyo University of Marine Science and Technology (TUMSAT), 4-5-7, Konan Minato-ku, Tokyo 108-8477, Japan

*Correspondence to*: Yukiko Nagai (nagai.y@jamstec.go.jp)



**Abstract.** Foraminifera are commonly used to reconstruct paleoenvironmental conditions based on the taxonomical composition, as well as elemental and/or isotopic signatures of their calcareous tests. A major problem, often referred to as the 'vital effect', is that such geochemical signatures stored in inorganic calcium carbonates differ greatly under the same

environmental conditions. This effect was previously explained by proportional contributions from passive vs active ion transport patterns, but their details are still investigated. In this study, the functional role of pseudopodial structures during chamber formation is elucidated by detailed observation of *Ammonia beccarii* (Linnaeus) using a time-lapse optical imaging system and high-resolution electron microscopy. For the first time, we document triple organic layers sandwiching carbonate precipitation sites. The three major organic layers (outer organic layer, primary organic sheet, and inner organic layer) are

formed by an initial framework of pseudopodia overlaid with further layer-like pseudopodia. The POS seems to facilitate early calcium carbonate nucleation, then entrapped by double precipitation sites. We further show that calcification starts when outer/inner organic layers still reveal tiny gaps (holes within the framework) that may serve as pathways for passive ion exchange (e.g., $Mg^{2+}$) between seawater and the confined precipitation space. Nevertheless, the majority of wall thickening occurs when the precipitation site is completely isolated from seawater that implicates of active ion exchange.

This may explain the differences in Mg/Ca ratios in early and later stages of calcification observed in previous studies. Our study resolves a key 'missing piece' in understanding foraminiferal calcification. The 'vital effect' is directly linked to spatio-temporal organization of the 'biomineralization sandwich' controlled by the three major organic layers. This study exemplifies the importance of culture experiments and in-depth observations of living organisms in order to interpret and calibrate biogeochemical proxies.

**1 Introduction**

Rotaliids are calcareous perforate foraminifera representing a group of marine protists classified within the Globothalamea class of the phylum Foraminifera (Pawlowski et al., 2013). They consist of well-established group of benthic and planktonic proxies. Rotaliid foraminiferal tests (shells) grow by additions of small compartments called 'chambers' sequentially with the growth of the cytoplasm (Haynes, 1981), each species has a characteristic test morphology. Foraminifera as a phylum

originated in the Precambrian and survived to modern days, they have experienced numerous diversifications and extinctions following global environmental changes. About 4,000 species have been described from the modern environment (e.g. Murray, 2007; Pawlowski *et al.*, 2014). Furthermore, approximately 50,000 to 100,000 species have been documented from the fossil record. These numerous foraminifera species have specific habitat and environment preferences and are limited to specific geological ages. Moreover, the test are easily fossilized after death and are preserved in the sediment through

geological time scales. The taxonomy and diversity of foraminiferal assemblages in each environment has been well





investigated throughout the geological age, and they are widely used as index fossils and facies indicators (e.g. Murray, 2007).

In recent years, the foraminiferal test has become widely applied as a palaeoenvironmental proxy, and its geochemical / isotopic composition has become one of the major tools in palaeoenvironmental reconstructions. The test morphology and

chemical composition to a certain extent depend on the environment (Schiebel *et al.*, 2017; de Nooijer *et al.*, 2014). The calcification process of the foraminiferal test is the phase of growth in which the elemental and isotopic compositions of the test is determined, and is also the key generating their morphological diversity. To this end, elucidating the detailed mechanisms of foraminiferal calcification has been treated with great interest in the field of geosciences. For example, it has been proven by experiment that the seawater temperature and the Mg/Ca ratio of foraminifera show a strong linear

correlation (Nürnberg *et al.*, 1996; Toyofuku *et al.*, 2000). Meanwhile, it is also known that the incorporation ratio Mg/Ca is variable and species specific (summary in Toyofuku *et al.*, 2011). The chemical distribution, however, vary among even individuals of the same species and exhibit zonation, corresponding to the test wall structure (Kunioka *et al.*, 2006; Van Dijk *et al.*, 2017). These variations in chemical composition, both inter- and intraspecific, are inclusively termed 'vital effect' (Urey, 1951). In order to reconstruct accurate palaeoenvironments, it is important to utilize reliable proxies, such as the

chemistry and isotopic composition of foraminifera tests. Therefore, the biological processes of chamber formation is of great importance and interest. Fortunately, since foraminifera still survive till modern days, it is possible to carry out *in situ* observations and design culture experiments for learning their biology and further improving palaeoenvironmental analysis. Despite this, the biomineralization process foraminifera is much less studied compared to that of bivalves and coccolithophores.

Observation of the foraminiferal chamber formation has been reported from as early as 1854 using the genus *Peneroplis* (Schultze, 1854), and many species have been documented thereafter (e.g., Myers, 1935, 1940, 1943; Jepps, 1942; Sliter, 1970; Berthold, 1976; Spindler and Rottger, 1973). Superfine structure observation by scanning and transmission electron microscopy (SEM and TEM) in order to carry out more detailed documentation of the cellular process of calcite precipitation during chamber formation in the benthic species *Rosalina floridana* (Angell, 1967) and the planktonic species *Globorotalia*

*truncatulinoides* (Hemleben *et al.*, 1986), *Orbulina universa* (Spero, 1988) have been reported. The common features summarized from these detailed observations on benthic and planktonic species, points to the fact that cytoplasm and the many types of organic sheet-like structures (i.e., organic layers like Outer Organic layer (OOL) and Inner Organic Layer (IOL)) play fundamental roles in calcification, as opposed to simple chemical reactions between calcium and carbonate ions. Pseudopodium is one of the key features of foraminiferal biology. Pseudopodia form a part of the cytoplasm consisting of

cytoskeleton structures, such as microtubules and actin filaments, as well as other organelles like mitochondria, vesicles, and vacuoles (Marszalek, 1969, reviewed in Travis and Bowser, 1991). Pseudopodium represents a multi-functional cellular structure serving various purposes such as locomotion, feeding, digestion, and chamber formation. Granuloreticulopodium (see Travis and Bowser, 1991) is included in this general term to define a granular reticulated pseudopodium responsible for feeding, digestion and locomotion. The appearance of pseudopodia changes during chamber formation and a fan-like array





of pseudopodia develops (Bé *et al.*, 1979). Then, an organic structure that forms the framework for chamber formation, called *Anlage*, is formed (Angell, 1967). In benthic foraminifers, an algal cyst composed of foreign detritus and other materials is constructed around this *Anlage* (Angell, 1967). *Anlage* is largely constructed by foamy and spherical microstructures (<1 μm) (Angel, 1967; Hemleben *et al.*, 1986), and is bulging in shape which led some authors to call it a

5 'bulge' in early studies using planktonic foraminifers (e.g., Bé *et al.*, 1979). This bulging *Anlage* is the three-dimensional structure that becomes the precursor of the chamber. There are three organic layers in the *Anlage*, one on the outer surface has been termed the 'Outer organic layer' (OOL; Spero, 1988), the one in the middle is named the 'Primary organic sheet' (POS) (Hemleben *et al.*, 1986; Erez, 2003), and the innermost one is called the 'Inner organic layer' (IOL; Spero, 1988). Precipitation of calcium carbonate microcrystals takes place on both sides of the POS, sandwiched between the Outer and

10 Inner organic layers. In addition to these three organic layers, the term *Anlage* is now loosely accepted to include the numerous pseudopodial cytoplasm that are present around them during calcification. Since different authors have different views and definitions as to what *Anlage* means (e.g., Angell, 1979; Bé *et al.*, 1979; Hemleben *et al.*, 1986), hereafter we refrain from using the term *Anlage* and instead use 'organic scaffolding' to refer to the organic framework which the chamber wall is built upon.

In order to investigate the fundamental functions of the POS, the Outer organic layer and the Inner organic layer during chamber formation, Nagai *et al* (2018) conducted focused ion-beam (FIB) processing on a foraminifera specimen during calcification, which allows the thin-sectioning of the site of calcification without decalcification to observe cytoplasm and the natural state of the calcifying test (calcium carbonate crystals) together using electron microscopy. Their observations clearly show that the organic scaffolding has numerous voids and empty spaces within the membranous structure of the site

of calcification (SOC). The presence of calcification liquid and exo/endocytosis are inferred, and the growth of calcium carbonate could be shown using time series samples. However, they have not documented the processes which leads to the construction of the POS and other organic structures during chamber formation.

Undoubtedly, the organic scaffolding built prior to chamber formation is an important factor shaping the characteristic morphology of foraminifera, serving as a template for calcification. When the foraminiferal test is dissolved, the organic

structure is revealed and it has the same overall morphology as the calcareous part (Banner and Williams, 1973). Despite it has been suggested that pseudopodial activity plays a key role in this process, little is known about the mechanism. Spindler and Röttiger (1973) first stated that it is pseudopodia that secrete the organic layer using optical microscopy, working with *Heterostegina depressa*. However, due to the low resolution of optical microscopy, they were unable to see the details of the process and this had no evidence other than largely speculation.

Although foraminifera are widely used for palaeoenvironment modelling, a total understanding of the foraminiferal calcification process is still lacking, impacting the accuracy of predictions made from foraminifera-based data. An accurate overview and model of the chamber formation by pseudopodia and the calcification process in calcareous foraminifera is therefore urgently needed to better our understanding of palaeoenvironments as well as predicting responses to ongoing climate change. To fill this knowledge gap, this study aims to elucidate the role of pseudopodial activities on the formation



process of foraminiferal chamber and its organic structures within the calcareous wall using the benthic hyaline foraminifera *Ammonia beccarii*, which has been used in a few relevant previous studies (e.g., Toyofuku *et al.*, 2017), as a model system. We combine differential interference contrast (DIC) microscopy and scanning electron microscopy, capturing DIC images through a time-lapse to document the pseudopodial activities during chamber growth and carried out SEM observations for

specimens fixed in different chamber formation process to visualize organic structures at the sub-micron order.

## 2 Materials and Methods

### 2.1 Sample Collection and Laboratory Culture

Living foraminifera were collected from brackish water salt marsh sediments of Hiragata Bay, Natsushima-cho Yokosuka, Japan (35°19'21''N, 139°38'5''E) in the spring of 2015. Surface (top 5 mm) sediments were collected and transported to the

10 laboratory to serve as a stock from which individuals of the benthic calcareous foraminifera *Ammonia beccarii* (*sensu* De Nooijer *et al*., 2008) were isolated. Living specimens were recognised by their bright yellow color and visible pseudopodial activity. They were cleaned from excess sediment and debris under a stereo microscope (SteREO Discovery V12, Zeiss Co. Ltd.) and transferred to filtered (0.2 μm) natural seawater (salinity ca. 35) and placed in a Petri dish. The Petri dishes were maintained at 20°C and twice a week, a small amount of live microalgae (*Dunaliella tertiolecta*, NIES-2258) were added.

Within a few days of feeding, some individuals started chamber formation and were selected for observation.

### 2.2 Optical Observation Settings of Chamber Formation

Chambers in the process of formation were observed using an inverted differential interference contrast (DIC) microscope (Axio Observer Z1, Zeiss, Germany). Time-lapse images were captured automatically by the digital microscope software Axiovision (Version 4.6). Time intervals between shots varied from 10 seconds to 10 minutes, but typically the interval was

20 1 minute. Magnifications of the available objective lenses were x10, x20 x40, and x63. A heat cut filter was applied to reduce damage on the living individuals inflicted by the image capture process.

### 2.3 Microstructure Observation and EDS Analysis

All specimens were fixed simultaneously using a fixing solution (3% paraformaldehyde, 0.3% glutaraldehyde, 2% NaCl in PBS buffer, pH 7.8) and subsequently stored in 2.5% glutaraldehyde at 4°C to avoid any morphological changes in the cell

material through dehydration. They were then washed in 0.2 μm filtered seawater, post-fixed with 2% osmium tetraoxide filtered seawater solution for 2 hours at 4°C. Following that the specimens were rinsed with distilled water and conductive staining was performed by incubating in 0.2% aqueous tannic acid (pH 6.8) for 30 minutes (Willingham and Rutherford, 1984). After another wash with distilled water, specimens were further treated with 1% aqueous osmium tetraoxide for 1 hour. Finally, they were dehydrated in a graded ethanol series and critical point dried (JCPD5; JEOL Ltd., Tokyo, Japan).

SEM observations were carried out on a JSM6700F field emission scanning electron microscope (FE-SEM) in Japan Agency



for Marine-Earth Science and Technology (JAMSTEC), Yokosuka, Japan. Elemental composition of all specimens was analysed using a JED 2300 (JEOL) dispersive spectrometer (EDS) equipped on the same JSM6700F FE-SEM at JAMSTEC. Selected specimens processed for SEM observation were embedded in epoxy resin for the purpose of measuring elemental composition of the newly forming chamber wall. The epoxy resin fully filled the chamber cavities and were polished to

expose the chamber wall being formed, and the exposed surface was coated with a ca. 3-nm-thick osmium foil. After rinsing with distilled water, this polished block was sectioned using an automicrotome to generate relief-free sections of foraminiferal tests revealing fresh calcite surfaces of chamber walls.

## 3 Results

We succeeded in observing the chamber formation in *Ammonia beccarii* using DIC and SEM techniques. Time-lapse

imaging with DIC observation was able to capture the process of chamber formation and the change in morphology over time at a micrometer order resolution, importantly also capturing the movement of cytoplasm and pseudopodia in detail. SEM observation revealed the fine submicron order processes leading to the construction of organic structures as well as the precipitation of calcium carbonate.

### 3.1 Time Series Observation with Optical Microscopy

We were able to observe the chamber formation process of *A. beccarii* for 59 times in total with DIC. Depending on the size of the chamber, it took about 5–8 hours to complete the whole process (Table 1). Prior to the start of chamber formation, exceptional activities were exhibited by the expanded pseudopodia. Usually for the purpose of feeding and moving, pseudopodia randomly branches at irregular intervals to arbitrary direction with variable lengths. During the chamber formation process, however, the pseudopodial activity significantly differed, in that a fan-shaped complex pseudopodial

network was constructed (Figure 1A), expanding from the aperture of the last existing calcified chamber. This pseudopodial network is arranged in a dense, radiating spray resembling that of a dandelion flowerhead. This unique characteristic allowed us to recognise individuals in the beginning of chamber formation and start our time-lapse observation from there (as 0 min). For an average individual, the events of chamber formation can be sequentially divided into three steps, outlined as follows in a typical time sequence.

The initial stage of chamber formation was from 0 to about 15 minutes. This is the stage where the organic framework for chamber formation is built. Following the pseudopodial network construction which takes place from 0 minute, an aggregation of cytoplasm quickly became visible around the aperture of the last existing calcified chamber (15 minutes; Figure 1B). As the cytoplasm expands, the pseudopodial network retracts. We consider the completion of the organic framework to be the end of the initial stage.

The middle stage of chamber formation took place at 15–60 minutes. During this stage, the foraminifer prepares the organic scaffolding for calcium carbonate precipitation which begins during this stage. By 30 minutes, the cytoplasmic aggregation concentrates in the same shape of a newly forming chamber like a hemi-sphere (Figure 1C). At this point, fine and short



pseudopodia have retracted to a certain extent but still seen on the surface of the structure. A brighter band of particles, probably representing calcium carbonate starting to become formed, can be seen on the surface of this. This proceeds to become the chamber wall. At around 60 minutes, the pseudopodia retracts beyond the forming chamber wall and the wall surface becomes smooth (Figure 1D).

The late stage of chamber formation is defined as the stage where calcium carbonate is precipitated extensively to thicken the chamber wall in the newly forming chamber, and takes places between around 60–400 minutes. We define the start of the late stage as when the pseudopodia beings to expand again to cover the organic scaffolding, and also the whole organic scaffolding is covered by a layer of calcium carbonate (with pores becoming visible under light microscopy). From around 60–100 minutes, the pseudopodia expand again to form a dense network, this time in thicker strands (Figure 2A). The length

of all the pseudopodia appear remarkably regular. Calcium carbonate continues to be precipitated in the forming wall. At this point, the overall outline and size of the newly forming chamber is basically fixed. Pseudopodial movement can be seen inside the forming chamber (open triangles in Figure 2A). Cytoplasm aggregate that filled the newly forming chamber retreats to the previously formed chamber. So empty space is made in the chamber. At 150 minutes, a network of pseudopodia is present in the forming chamber, the chamber wall of which thickens and the pores become increasingly and

clearly visible (from Figure 2B–C). Chamber thickening continues to occur from 150–400 minutes (Figure 2C). During this process, the density of the pseudopodial network on the chamber wall surface is increased and wraps the chamber wall like a mesh. As the chamber wall thickening completes at around 400 min, the mesh-like pseudopodial network on the surface disappears (Figure 2D). We consider this to indicate the termination of chamber formation process. After this, the individual starts to show the usual type of pseudopodia movement.

**3.2 Ultramicro Observations on the Forming Chamber Wall**

The process of chamber formation is classified into three stages, as outlined above. Specimens exemplary of each stage were observed with a scanning electron microscope. A schematic diagram is presented in Figure 3, which outlines the general observations. The basis of organic layer formation is the interweaving of a pseudopodial framework (Figure 3A), the interspaces of which is then filled in with a further layer of pseudopodial material, resulting in a complete organic layer. The

25 pseudopodia are observed to form a dense framework (purple dotted lines in Figures 2A, 3 and 4), which is then overlaid by a layer of membranous pseudopodia which fills the interspaces (Figure 4E). In the OOL, numerous spaces of 100 nm – 1 um can be seen (gray in Figures 2–4), which represents the interspaces between the framework which is yet to be filled. In some instances, the membranous pseudopodia were observed during the process of filling the interspaces, sometimes from more than one direction (e.g., Figure 4E), by a gradual, webbed expansion.

**Initial Stage**

In the initial stage of chamber formation, the test was entirely covered with pseudopodia and organic layer-like structures (Figure 4A), some parts of these covering structures were peeled off during the sample preparation process. Focusing on the



chamber being formed, it was possible to observe the OOL and the POS (Figure 4B), with the POS being visible from gaps in the OOL. The interspace between the two layers was narrow (Figures 4B-C). Even at high magnification, the outer surface of the OOL itself is relatively smooth layer-like structure (OOL in Figure 4B-C). In some cases, the pseudopodia can be seen directly expanding from the OOL (Figure 4C). The primary organic sheet (POS) can also be observed (Figure 4B) and is

relatively robust, covered by numerous protrusions. These are convex, frustoconical structures about 1 μm in width (Figure 4A-B), and represent pore plates which corresponds to pores. These are simultaneously formed when the POS was constructed (green coloured in Figure 4C-D). Projections (<1 μm) were observed on the cytoplasmic surface of OOL (light green in Figure 4B). Vesicles can be seen on the OOL (blue coloured in Figure 4C, some appeared crushed probably due to the critical point drying process), and similar structures could also be found on the POS (blue coloured in Figure 4D and E).

The size of vesicles varied from 50 nm to 500 nm, and these likely represent vesicles. On the OOL, some pseudopodia appeared to have a form like that of a sausage chain (Figure 4B), the diameter and interval of contractions were variable. The bulging part contained only cavities and this form might be associated with peristalsis. It is known pseudopodia transport mitochondria and vesicles (Travis and Bowser, 1991; Cedhagen and Frimanson, 2002), and it is possible that this peristaltic structure has important roles in such transportation.

No crystals were found between the POS and the OOL, indicating that no calcium carbonate has been deposited at this stage, supported by the fact that the SEM-EDS analyses showed an absence of calcium signals (Figure 7A).

**Middle Stage**

At the middle stage, the interspaces among the framework structure constructed by the pseudopodia has been filled to a

much larger extent than in the initial stage, with much fewer gaps (about 5 nm – 200 nm; grey coloured in Figure 5B and C) that could be seen. Nevertheless, calcium carbonate precipitation has already started between organic layers in some parts of the forming chamber (Figure 5D). Upon closer observation, these were revealed to consist of needle-like structures that covered the surface of the POS, close to the previously formed chamber. These needle-like structures were confirmed to be crystals of calcium carbonate precipitating vertically between the OOL and the POS by EDS observation (Figure 7B).

Therefore, the precipitation does not start at the same time across the entire chamber, but instead begins locally right after the completion of organic layer construction. At this point, there are still small gaps between independent crystals. It can also be noted that the framework structure formed by pseudopodia appears to have a certain directionality in growth.

Numerous, rather regularly spaced pores (about 1 μm) can be clearly observed on the crystalline layer (Figure 5A). In the part where OOL was curled up to reveal the inner side (see Figure 3B), convex structures corresponding to pore lining were

seen (Figure 5D). This has been termed 'pore funnel' by Hottinger (2006), which we adopt here. Interestingly, pores cannot be seen at this stage from the outer side on the OOL with SEM observation (Figure 5A-B), and the OOL appears entirely smooth in the parts where the framework has been filled completely. We interpret this as due to a layer of cytoplasmic material also fills the pore lining (i.e., the 'well') during chamber formation, which regresses after the completion of chamber formation (and therefore becomes visible under SEM). As discussed previously, however, pores can still be seen





during chamber formation using light microscopy due to the semi-transparent nature of the organic layers as well as the thin calcium carbonate layer. Algal cysts including *Dunaliella* individuals can be seen (Figure 5A). The OOL is a continuous structure that envelopes the entire test, and it extends to the newly forming chamber from the aperture of the previously formed chamber. In some parts where calcium carbonate precipitation has not yet taken place, the outer surface of the POS

can be seen (Figure 5A and 5C) and like in the initial stage, many frustoconical structures about 1 μm in width are seen (Figure 5C). Vesicles (blue coloured in Figure 5B-C), about 50 nm – 500 nm in size, could be seen on both the OOL and the POS as in the initial period.

**Late Stage**

In this final stage (Figure 6), the construction of organic layers has been fully completed, and a layer of calcium carbonate began precipitation across the entire forming chamber. The OOL is therefore seen as uniformly smooth and without gaps from the outer side (Figure 6B). Cross-section through the forming chamber wall at the late stage clearly shows three completed layers (corresponding to the IOL, the POS, and the OOL from the inner side outwards in that order) and two layers of precipitating calcium carbonate sandwiched between the IOL and the POS as well as between the POS and the

OOL (Figure 6C). EDS analyses obtaining signals of Ca and C, O simultaneously (Figure 7C) clearly indicated high Ca signal distribution being detected these two layers, showing that these layers are calcium carbonate in nature.

The precipitation of calcium carbonate crystals, continuing from the middle stage, leads to carbonate crystals to become increasingly densely packed, with gaps between crystals completely disappearing by the end of the late stage (which marks the end of chamber formation). In the figured specimen observed in Figure 5, the thickness of the calcium carbonate

layer is about 1 μm between the OOL and the POS, and about 0.3 μm between the POS and the IOL. OOL had toward the inner side (Figure 5D).

As in the middle stage the exterior of the OOL appears smooth (i.e., pores cannot be seen yet) (Figure 5D). The IOL, however, when seen from the cytoplasm side, is seen to be covered by regular depressions that corresponds to the convex side of the pore plate on the POS (which may be named the 'inner pore'). The IOL can therefore be considered to have the

25 same shape as the POS. Vesicle-like structures could also be observed in the late stage on the surface of the OOL but the size of these structures was more variable than in the earlier stages, ranging from 50 nm to 1 μm (Figure 5C-D). Furthermore, similar structures could also be observed on the IOL (not shown).

## 4 Discussion

### 4.1 The Weaving of Organic Layers During Chamber Formation

This study is the first to observe the detailed making of organic layer during chamber formation, and revealed that the layers are actually woven by pseudopodial activity. Initially, a framework is constructed by a pseudopodial network, which is then overlaid and the interspaces filled in by a layer of membranous pseudopodia. The importance of organic layers in the early



stages of chamber formation has been speculated in the previous study, but little was known about its origin. It was previously thought that the organic layer was secreted from the pseudopodia (e.g., Angell, 1967; Röttiger, 1974; Hemleben *et al.*, 1986), and Spindler and Röttiger (1973) reported that the organic layer seems to be connected with pseudopodia. These studies were largely limited in that their magnification (only light microscopy was available then) was not sufficient

resolution to observe the detailed process. The process documented herein provides evidence for an entirely novel model in that the pseudopodia itself weaves the organic layers – in other words the organic layer is the part of cytoplasm.

### 4.2 Pore Formation

Fine-scale observations from the present study allowed us to reconstruct the actual steps in pore formation. As shown already in previous studies (Bé *et al.*, 1979; Spero, 1988), the structure known as 'pore' in foraminifera is actually a

composite structure formed by two opposing wells converging at the POS, one opening towards the outer side located on the OOL and one opening towards the cytoplasm side located on the POS (and same on the IOL). The POS/IOL well has been called the pore plate in previous studies (e.g., Haynes, 1981). These pore plates can also be seen on the organic layer template when fossil foraminiferal tests are dissolved (Bannar *et al.*, 1973; Banner and Williams, 1973; Hottinger and Dreher, 1974; Cader *et al.*, 2003; Ni Fhlaithearta *et al.*, 2013). Therefore, the pores are not actually pass-through structures formed at

once but are instead formed in unison by separate processes on the OOL and the IOL. Our observations show that in the initial stage of chamber formation, the pore plate (visible as frustoconical structures of about 1 μm) is already present when the POS is woven, at the growth front. Pore funnels, about 0.5 μm in size, that pair up with the pore plate in the same location (but open to the opposite direction) are formed on the OOL. This structure and the pore plate collectively form the pore, and there is no space between the two for calcium carbonate to precipitate, and therefore the pore is not calcified. All

hyaline foraminifera that have been observed in detail possess pores. Since pores are not pass-through and formed as the framework for organic layer (i.e., OOL, POS, and IOL) formation is woven and that the layers are somewhat flexible before calcification, one possible speculative function for pores is to serve as a connective structure between OOL and IOL. In this scenario, the pores 'staple' the organic layers of the forming chamber together, so that the sites of calcification maintain a consistent thickness and form throughout the chamber while calcification occurs.

**4.3 Vesicles**

The existence of vesicles on the surface of organic layers have been reported in previous studies (Angell, 1967; Spero, 1988), but their function and significance have not been mentioned. A recent study (Nagai *et al.*, 2018) that utilized Focused Ion Beam (FIB) technology to process SEM samples in order to visualize calcium carbonate and organic layers on the same semi-thin section. They were able to observe the presence of vesicles in the site of calcification, and that they might be

responsible for exo- and endocytosis. The vesicles increase the surface area and probably serve to improve the material exchange efficiency, by increasing the contact surface area with seawater. In the present study, we could observe numerous vesicles on all three organic layers, including the OOL, the POS, and the IOL. This indicates that the vesicles probably play





important roles in material exchange during calcification for both the outer and inner calcified layers, and as the vesicles are inferred to result from the activity of the organic layers this further strengthens the active role of these layers in calcification (i.e., they are not mere templates).

### 4.4 Prospects for Calcification Model

Until now, the exact process of calcium carbonate precipitation, in terms of how precipitation was related to the degree of isolation of the site of calcification, remained largely unclear (Erez, 2003; De Nooijer *et al.*, 2014). In the present study, the sequence of events during calcification was made clear by time-series observations, and importantly both the formation of organic layer and calcium carbonate precipitation were observed together. It is significant that during the Middle Stage, although the overall shape of the forming chamber has already been formed by framework-like pseudopodia, the

precipitation of calcium carbonate was seen to initially start before the framework pseudopodia have been fully covered and filled by membranous pseudopodia. The organic layers (especially well-observed in the OOL and the POS) still contained numerous gap <1 μm in size, which we interpret to maintain the exchangeability of seawater and elements contained within, for the initial part of calcium carbonate precipitation. The site of calcification is therefore interpreted to be still open in the Middle Stage. In the Late Stage, however, the organic layers have been completely filled by membranous pseudopodia and

no such gaps remain. At this stage, therefore, the site of calcification is closed from the surrounding seawater. Hence, we interpret that during the Late Stage the elements require for calcification must be selectively taken up by biological means such as exo-endocytosis or ion pumps through the OOL. Although we could not observe the IOL in detail (due to its position below the POS) during this process, the IOL most likely receives the required elements through pseudopodial transport during the Late Stage, although whether this originate directly from the forming chamber or the previous chambers cannot be

ascertained yet. Previous evidences (e.g., Toyofuku *et al.*, 2008; De Nooijer *et al.*, 2009) appear to suggest that calcium and carbonate are transferred from the previously formed chamber. The POS has been widely considered to be the only template for calcification (Hemleben *et al.*, 1986), but recent research has revealed that calcium carbonate precipitation also occurs on the other organic layers (Nagai *et al.*, 2018). It was also shown that the POS gradually becomes obsolete as the chamber matures towards completion of thickening. Therefore, the true role played by the POS during calcification should be

reconsidered. A likely function of the POS is that by doubling the surface area on which precipitation occurs, the existence of the POS doubles the rate of chamber formation. Considering that the mobility of foraminifera is highly limited during chamber formation, increasing the efficiency of chamber formation is probably beneficial and adaptive to the foraminifera.

It is well known that the chemical and isotopic compositions of calcareous foraminifera tests differ significantly from those

precipitated inorganically, and the compositions also differ among different species. This effect is collectively known as the 'vital effect' (Urey *et al.*, 1951), and has been a great hindrance to the use of foraminifera tests as geochemical proxies, for example to reconstruct palaeoclimates. In attempt to explain the vital effect, Nehrke *et al.* (2013) proposed a transmembrane transfer / passive transfer (TMT/PT) model by observing Mg/Ca ratio during calcification, assuming that low ratio indicates




active transport (i.e., transmembrane transfer *sensu* Nehrke *et al.*, 2013) and high ratio indicates passive transport; as Mg is discriminate against in Ca channels in active transport. Their observations indicated that that passive transport predominates at the early period of calcification, with active transport becoming dominant at later periods. This is consistent with the results outlined above from our observations during the present study, but we were able to reveal the reasons behind the differences in Mg/Ca ratios in early and later periods of calcification, which is that during the Middle Stage the site of calcification has not yet been fully enclosed. This is a key piece of finding as to what actually causes the vital effect, in that the construction process of the organic layers can significantly influence when the site of calcification becomes isolated, leading to differences in chemical and isotopic compositions of the test by the proportion of contributions from passive vs active transport.

## 5 Conclusion

Calcareous foraminifera are a highly important group in palaeoclimate reconstruction and as indication fossils, by using their chemical and isotopic composition as a geochemical proxy. A major problem was that such compositions differed greatly from inorganic calcium carbonate under the same environment. The key finding of the present study is that one main contributor to this 'vital effect' is in fact the proportion of contributions from passive vs active transport in material transfer during calcification, which is directly linked to how the three major organic layers (i.e., the OOL, the POS, and the IOL) are constructed. For the first time, this study revealed that the organic layers are in fact woven by a framework-like pseudopodia network that are then overlaid by an overlaying layer of membranous pseudopodia, closing the gaps in the framework and thus forming a complete organic layer. We show that calcification has already started when the site of calcification is still able to passively exchange elements (e.g., Mg) with seawater; but the majority of wall thickening occurs when it is completely isolated and the only means of element exchange is through active transport. This agrees with and explains the differences in Mg/Ca ratios in early and later periods of calcification observed in previous studies (e.g., Nehrke *et al.*, 2013). As such, we resolved a key 'missing piece' in understanding foraminiferal calcification that has mystified us for more than a decade. This study exemplifies the importance of extensive rearing and in-depth observations of a living species in order to correctly use biominerals as a geochemical proxy.

**Author Contributions**

Scientific conception and experimental design: YN and TT. Data acquisition and analysis: YN, TT and KU. Data processing: YN. Data interpretation: YN, TT, CC, JT. Manuscript writing and editing: YN, TT, CC, KU, RW and JT. YN and TT contributed equally to this work.





**Competing interests**

The authors declare that they have no conflict of interest.

**Acknowledgments**

The authors thank Nanami Kishigami, Sunaho Kubo, Yuki Iwadate, Sachiko Kawada (JAMSTEC), and Shunzo Kondo (JEOL) for their technical assistance and scientific advice on this study. This work was supported by a grant from the Faculty of Environment and Information Sciences, Yokohama National University (to YN) and a JSPS KAKENHI Grant Number 25247085 (to TT). JT received support from the Polish National Science Centre (UMO-2015/19/B/ST10/01944).

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

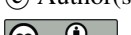


*Table 1. List of observation using inverted differential interference contrast microscopy*

| Date (YYYYMMDD) | Length of forming chamber (μm) | Width of forming chamber (μm) | Observation time (min) | Test diameter after chamber formation (μm) | Recorded stages of chamber formation |
|---|---|---|---|---|---|
| 20140401 | 108.12 | 42.24 | 300 | 248.05 | Middle-last |
| 20140402 | 151.45 | 56.31 | 60 | 282.63 | Initial stage only |
| 20140404 | 61.34 | 50.22 | 270 | 135.68 | Middle stage-last |
| 20140407 | 100.58 | 50.4 | 400 | 210.23 | Initial satge-last |
| 20140408 | 104.26 | 58.3 | 395 | 218.69 | Middle stage-last |
| 20140416 | 119.28 | 61.61 | 385 | 287.01 | Middle stage-last |
| 20140417 | 84.61 | 48.45 | 385 | 163.12 | Middle stage-last |
| 20140422 | 127.53 | 83.74 | 465 | 318.79 | Middle stage-last |
| 20140430 | 88.42 | 56.5 | ND | ND | Initial stage only |
| 20140430 | 88.42 | 56.5 | 345 | ND | Middle stage-last |
| 20140509 | 165.95 | 65.63 | 342 | ND | Middle stage-last |
| 20140522 | 94.98 | 57.48 | 360 | ND | Initial satge-last |
| 20140523 | 82.71 | 65.64 | 380 | ND | Initial satge-last |
| 20140711 | 72.21 | 48.94 | 138 | 141.63 | Late stage-last |
| 20140718 | 112.04 | 62.59 | 336 | 284.88 | Initial stage-last |
| 20140808 | 70.56 | 45.53 | 303 | ND | Initial satge-last |
| 20140822 | 96 | 59.13 | 315 | ND | Initial satge-last |
| 20150410 | 99.89 | 45.64 | 303 | 209.47 | Initial stage-last |
| 20150421 | 96.39 | 28.91 | 396 | 178.38 | Initial satge-last |
| 20150817 | 109.14 | 53.03 | 390 | 234.97 | Initial satge-last |
| 20150820 | 146.44 | 94.87 | 475 | 539.68 | Middle stage-last |
| 20150821 | 188.2 | 102.16 | 405 | 517.24 | Middle stage-last |
| 20150903 | 95.77 | 45.81 | 295 | 266.01 | Middle stage-last |
| 20150904 | 96.8 | 58.19 | 360 | 284.16 | Middle stage-last |
| 20150911 | 103.06 | 41.26 | 125 | 291.85 | Late stage-last |
| 20150917 | 92.81 | 55.55 | 325 | 199.63 | Middle stage-last |
| 20150924 | 93.61 | 51.28 | 345 | 204.19 | Middle stage-last |
| 20151001 | 92.08 | 47.51 | 345 | 312.03 | Middle stage-last |
| 20151014 | 79.14 | 58.7 | 380 | 193.5 | Middle stage-last |
| 20151015 | 81.64 | 55.47 | 370 | 175.45 | Middle stage-last |
| 20151111 | 142.81 | 98.42 | 420 | 361.39 | Middle stage-last |
| 20151118 | 84.96 | 54.82 | 270 | 210.6 | Late stage-last |
| 20151119 | 88.66 | 53.6 | 310 | 218.69 | Middle stage-last |
| 20151120 | 186.12 | 118.27 | 550 | 448.84 | Middle stage-last |
| 20151202 | 102.52 | 57.23 | 380 | ND | Middle stage-last |
| 20160127 | 110.04 | 67.85 | 355 | 258 | Middle stage-last |
| 20160302 | 110.47 | 62.45 | 20 | ND | Late stage-last |
| 20160610 | 121.37 | 39.38 | 325 | 317.71 | Middle stage-last |
| 20160611 | 99.89 | 57.42 | 420 | 206.43 | Middle stage-last |
| 20160612 | 107.65 | 27.7 | 335 | 253.74 | Middle stage-last |
| 20160613 | 132.45 | 61.4 | 140 | 317.63 | Late stage-last |
| 20160614-1 | 102.86 | 54.75 | 325 | 245.19 | Middle stage-last |
| 20160614-2 | 99.83 | 41.69 | 150 | 284.98 | Late stage-last |
| 20160618 | 132.4 | 52.5 | 330 | 288.37 | Middle stage-last |
| 20160619 | 138.03 | 34.84 | 270 | 238.39 | Late stage-last |
| 20160621 | 122.66 | 71.06 | 160 | 277.31 | Late stage-last |
| 20160622 | 101.2 | 40.52 | 465 | 240.3 | Middle stage-last |
| 20160623 | 100.34 | 44.88 | 105 | 259.44 | Late stage-last |
| 20160624 | 84.8 | 50.3 | 60 | 233.67 | Late stage-last |
| 20160929 | 114.94 | 62.78 | 214 | ND | Late stage-last |
| 20170221 | 119.67 | 77.47 | 57 | ND | Late stage-last |
| 20170310 | 106.65 | 65.62 | 21 | 277.45 | Late stage-last |
| 20171201 | 83.14 | 45.8 | 459 | 199.71 | Middle stage-last |
| 20171202 | 151.85 | 97.62 | 50 | ND | Late stage-last |
| 20171206 | 120.69 | 61.47 | 582 | 287.28 | Middle stage-last |
| 20171207 | 77.9 | 55.65 | 248 | ND | Initial satge-last |
| 20180104 | 90.5 | 38.1 | 420 | ND | Middle stage-last |
| 20180105 | 114.3 | 54.3 | 377 | ND | Middle stage-last |



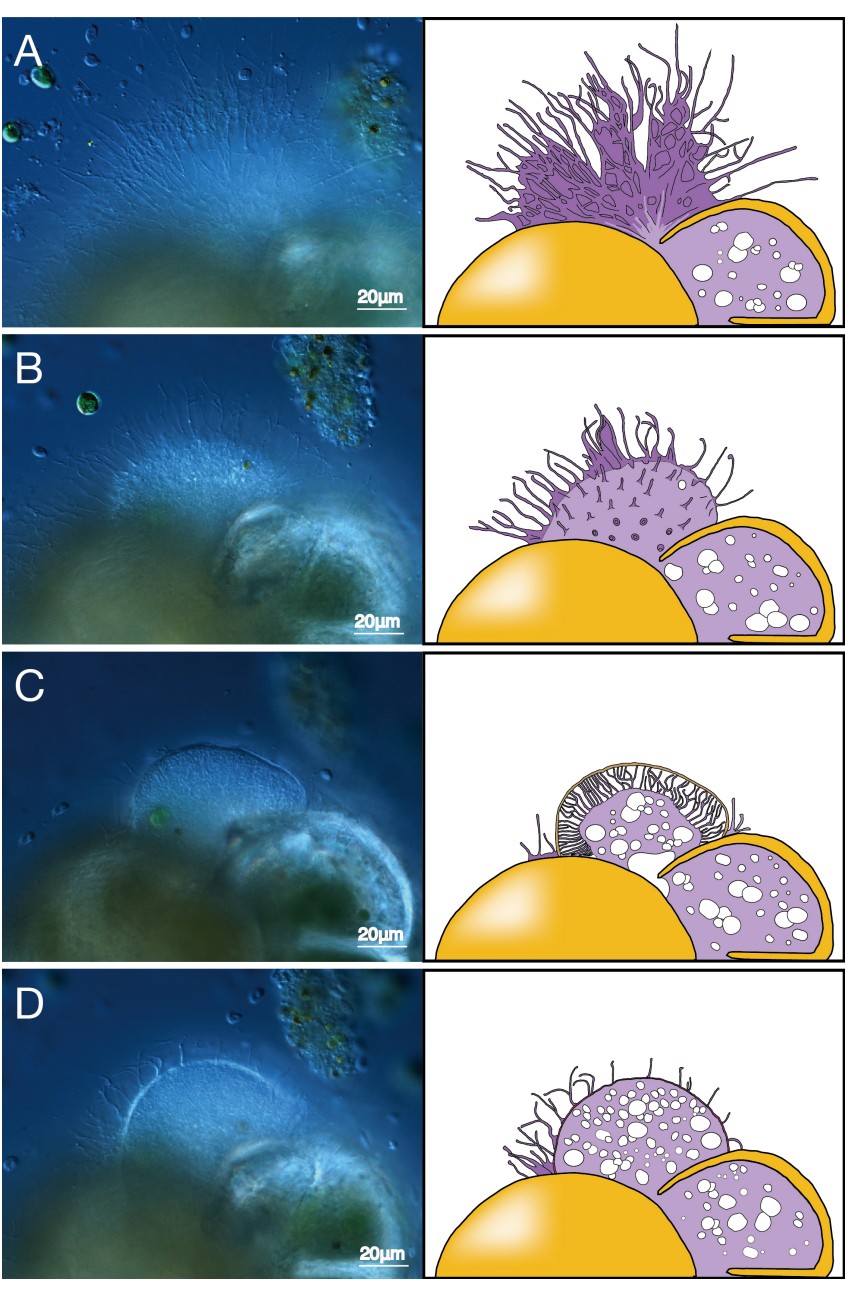





Figure 1: Time series observation of chamber formation by optical microscopy. A: Beginning of chamber formation, defined as 0 minute from the start. B: 15 minutes. C: 30 minutes. D: 60 minutes. Open triangles indicate pseudopodia inside the newly forming chamber. Left: optical microscopy image. Right: the same image with schematic overlay; colour legend: deep purple = pseudopodia; light purple = cytoplasm; magenta = calcium carbonate in the newly forming chamber; yellow = previously formed chambers.





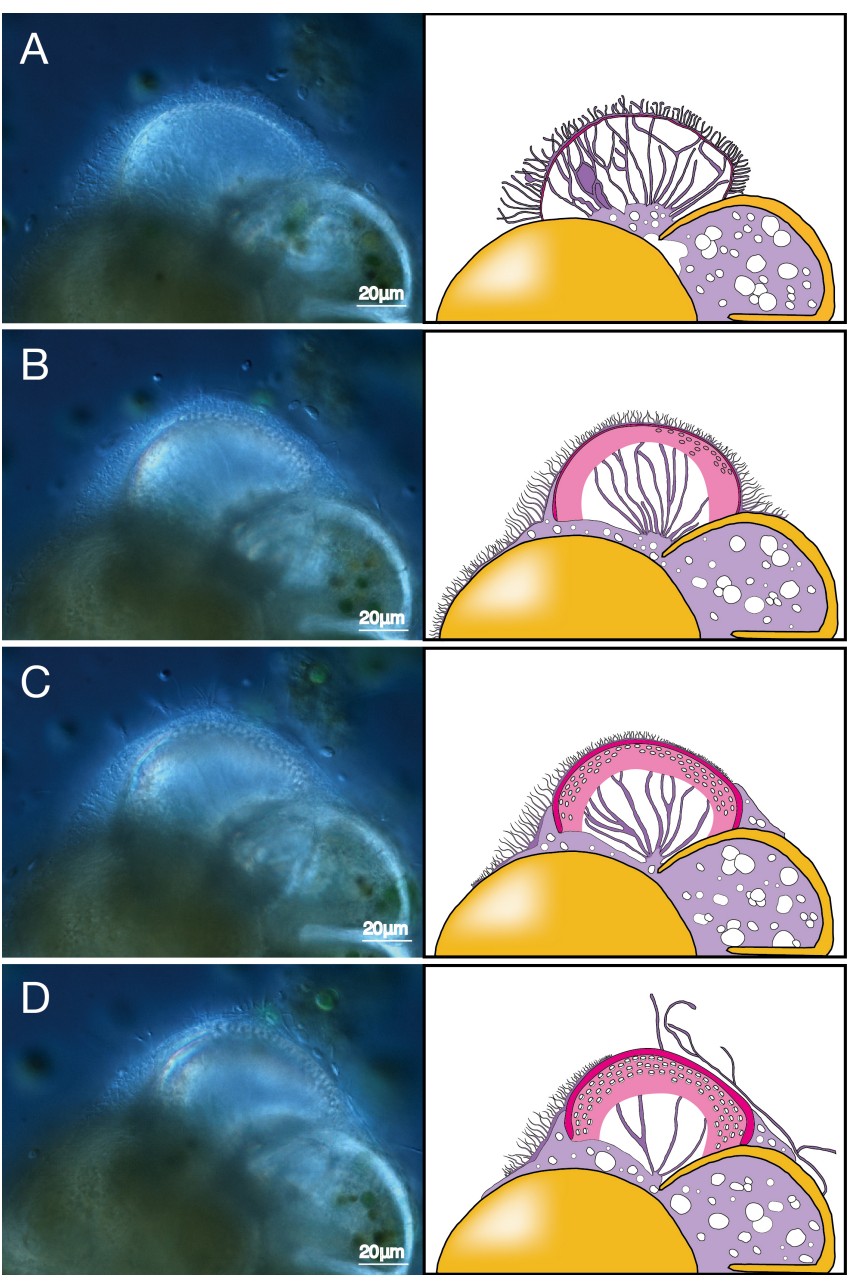





**Figure 2: Time series observation of chamber formation by optical microscopy (continued). A: 100 minutes. B: 150 minutes. C: 200 minutes. D: 400 minutes. Open triangles indicate pseudopodia inside the newly forming chamber. Left: optical microscopy image. Right: the same image with schematic overlay; colour legend: deep purple = pseudopodia; light purple = cytoplasm; magenta = calcium carbonate in the newly forming chamber; yellow = previously formed chambers.**

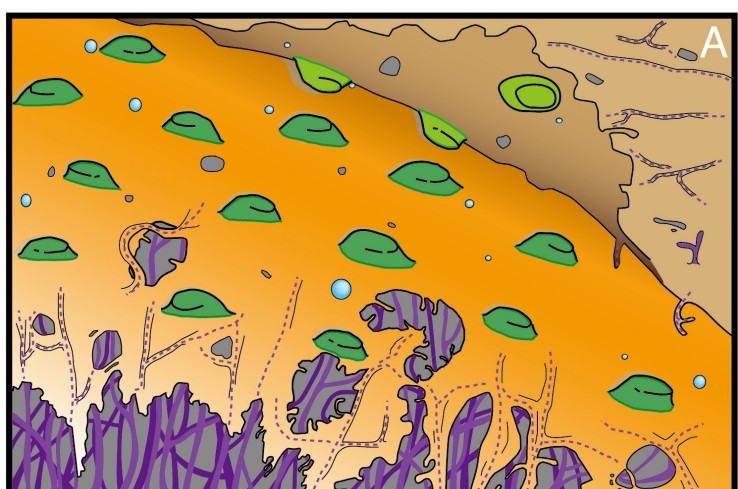

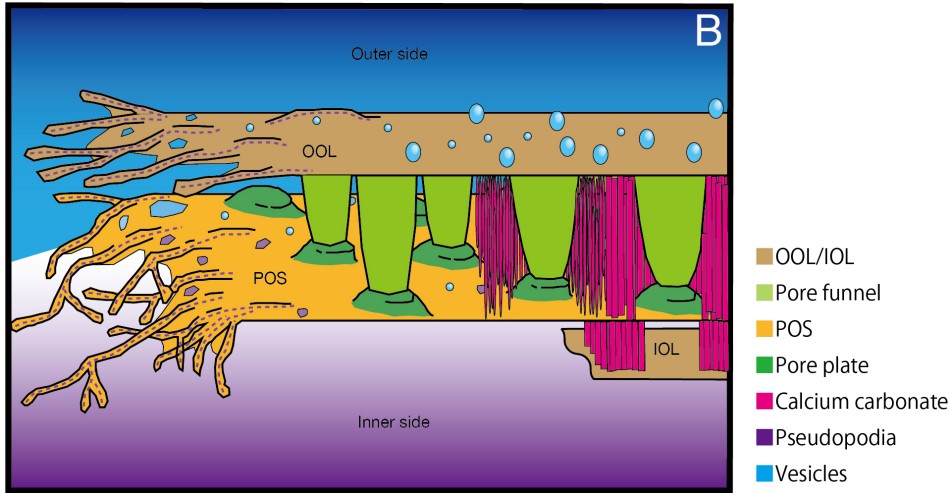

**Figure 3: Schematic illustrations of chamber formation. A: Construction of organic layers by pseudopodial weaving and subsequent gap-filling. B: The entire chamber formation process from the initial stage on the left side to the late stage on the right side. Colour legend: brown = OOL /IOL; orange =POS, purple = pseudopodia/cytoplasm; light green = pore funnel on the OOL; green = pore plate; magenta = calcium carbonate; blue = vesicles; gray =gap.**



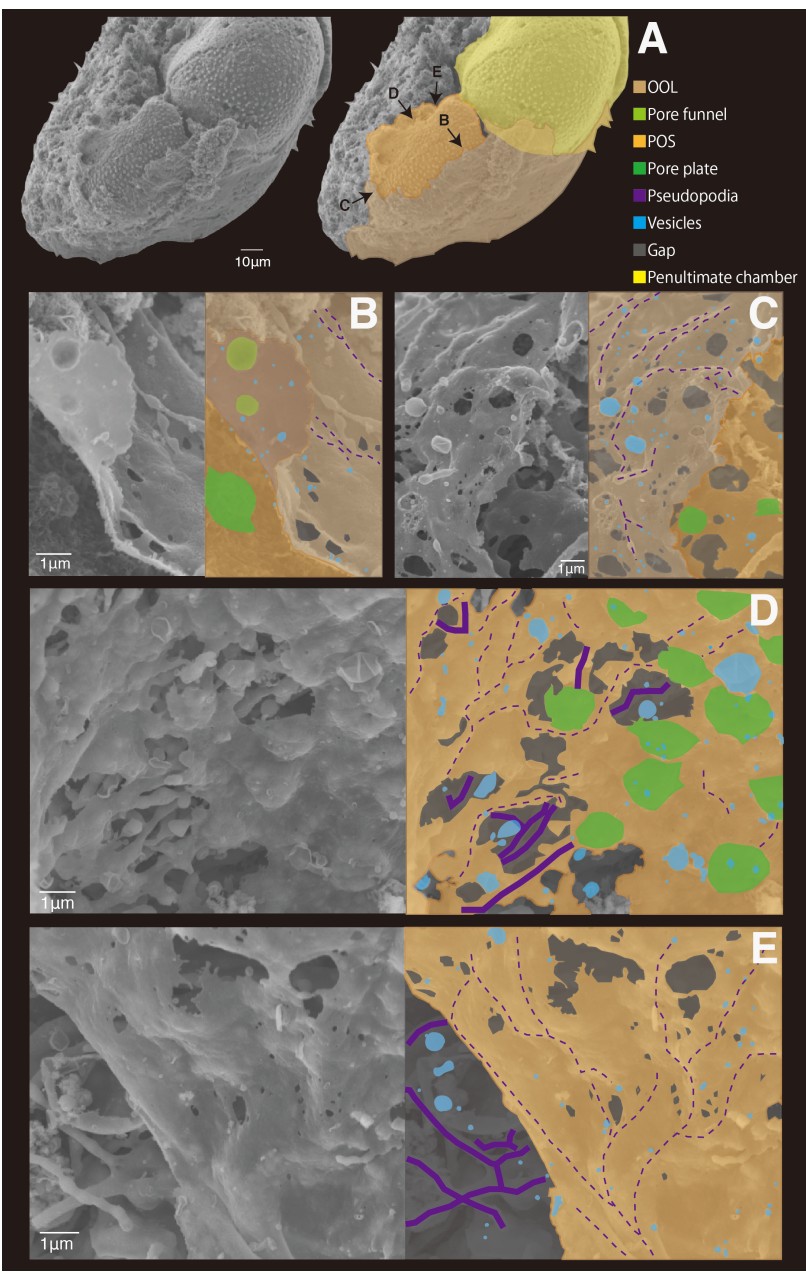



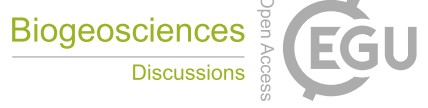

**Figure 4: Microstructures during the initial stage of the chamber formation shown by SEM images on the left, supplemented by schematic explanation on the right. A: Overview of a specimen showing the OOL covering both the newly forming and older chambers. B–C: Magnified images showing the OOL and the POS. D–E: Magnified image of the POS construction front showing the weaving action of pseudopodia. E: The same POS construction front showing the membranous pseudopodia extending so as to**

5     **close a large hole (white arrow). Colour legend: brown = OOL; orange = POS; purple = pseudopodia/cytoplasm; light green = pore funnel on the OOL; green = pore plate; blue = vesicles; gray =gap.**



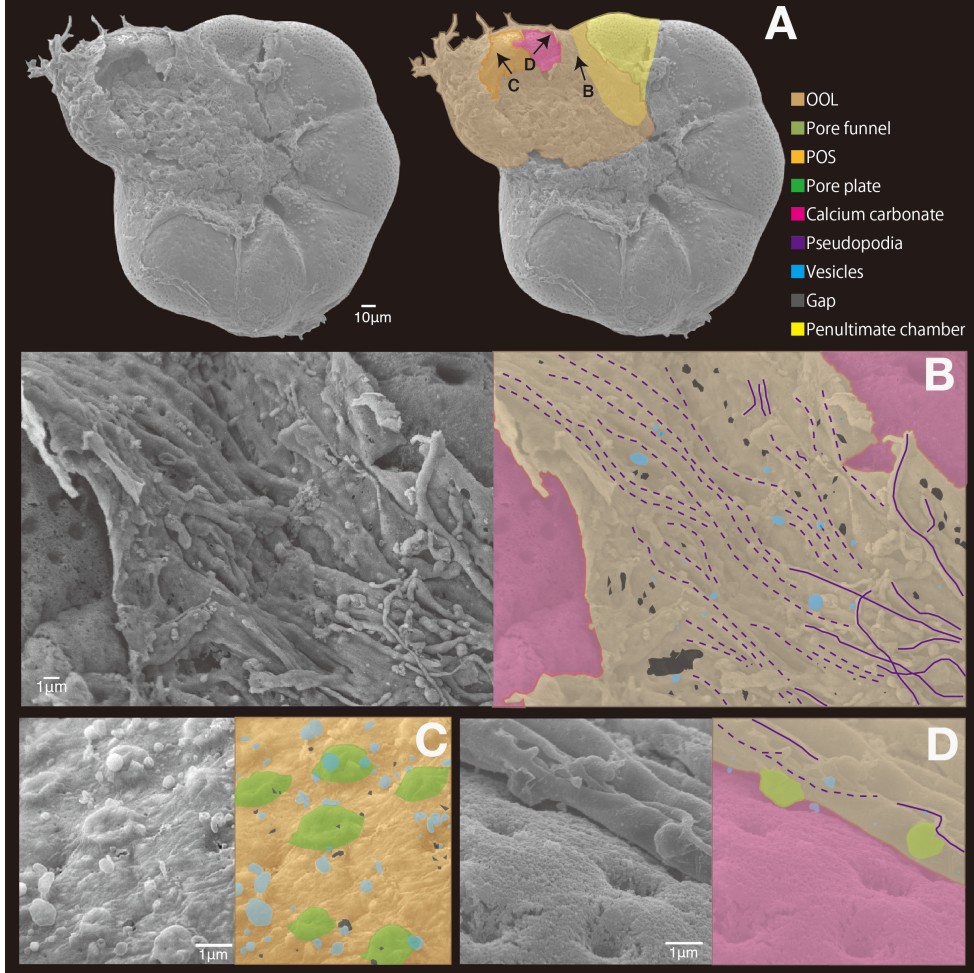

**Figure 5: Microstructures during the middle stage of the chamber formation shown by SEM images on the left, supplemented by schematic explanation on the right. A: Overview of the ventral side of a specimen, showing the cytoplasm covering the newly**

5 **forming chamber. B: Magnified image showing the OOL on the suture, between the new chamber and the previous chamber. C: A higher magnification image of the POS showing spherical structures on the POS. D: Image showing the matching relationship between convex structures on the cytoplasmic surface of the OOL and the pore. Colour legend: brown = OOL; orange = POS; purple = pseudopodia/cytoplasm; light green = pore funnel on the OOL; green = pore plate; magenta = calcium carbonate; blue = vesicles; gray =gap.**



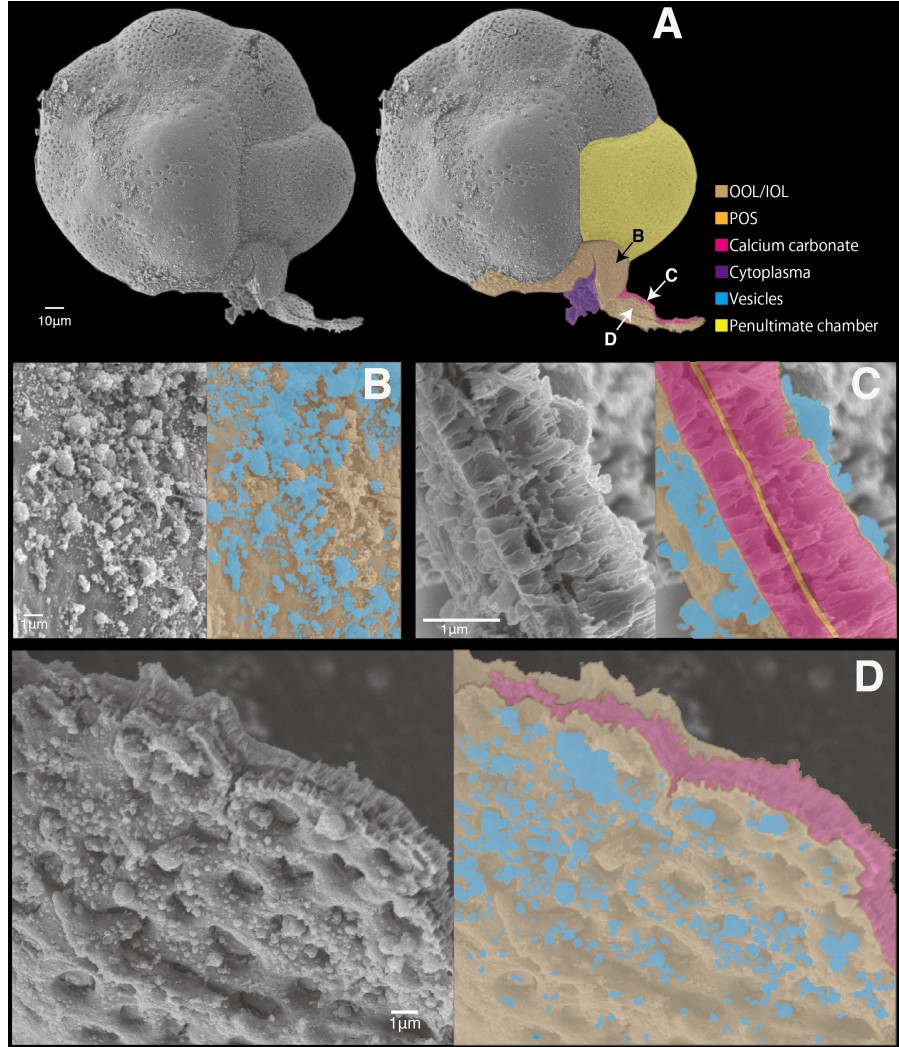

**Figure 6: Microstructures during the late stage of the chamber formation shown by SEM images on the left, supplemented by schematic explanation on the right. A: Overview of the dorsal side of a specimen, with the newly forming chamber on the bottom. B: Magnified image of the OOL seen from the outside. C: Image showing a cross-section through the forming chamber wall. D: A**
5 **magnification of the IOL seen from the inner side, showing pores and lots of spherical structures. Colour legend: brown = OOL/IOL; orange = POS; purple = pseudopodia/cytoplasm; light green = pore funnel on the OOL; green = pore plate; magenta = calcium carbonate; blue = vesicles.**



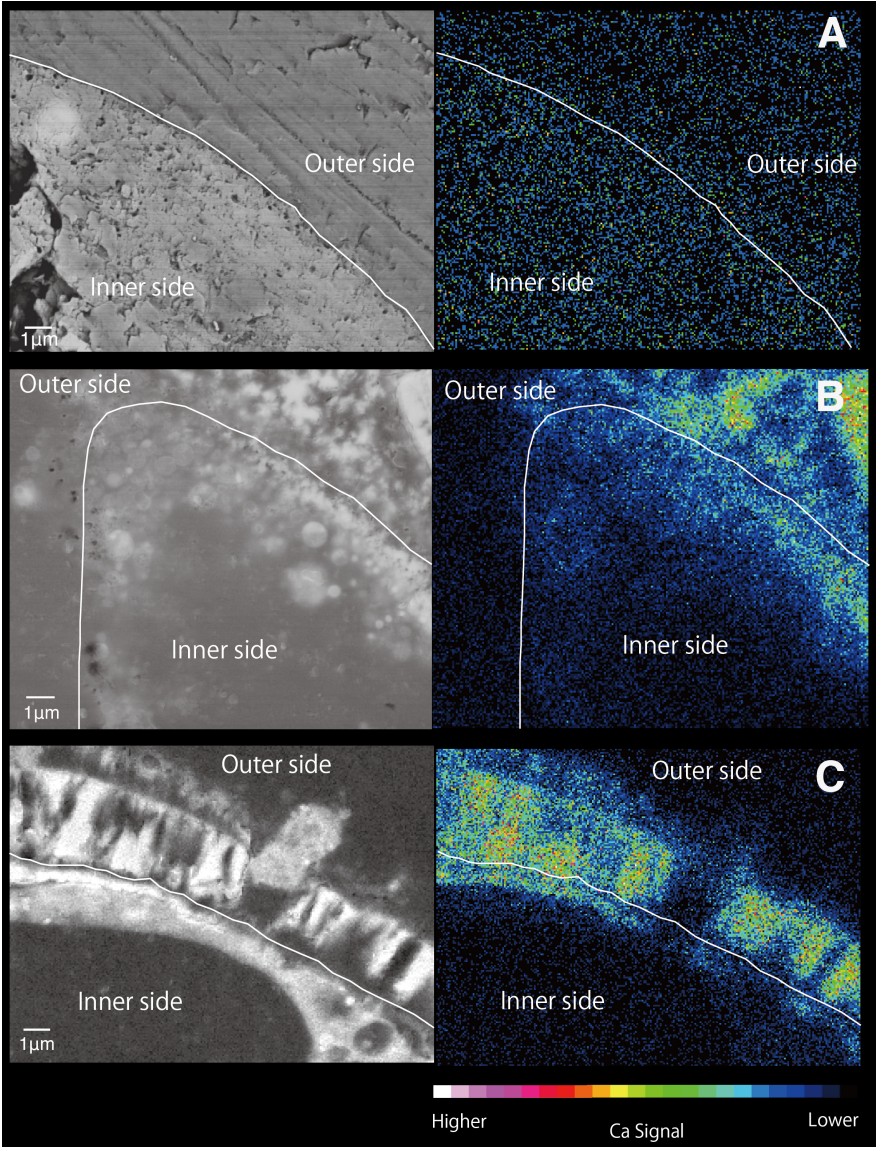

**Figure 7: Elemental maps of cross-sections through the forming chamber wall at different stages, shown by SEM-EDS analyses. A: Initial stage. B: Middle stage. C: Late stage. White lines indicate the position of the POS. The false color maps indicate the intensity of calcium signals, corresponding to the legend shown on the bottom.**