# Peer review of "Weaving of biomineralization framework in rotaliid foraminifera: Implications for paleoenvironmental reconstructions"

_Biogeosciences, 2018_

## Referee Comment (RC1) · I. van Dijk (Referee) · 7 Aug 2018

The study by Nagai and co autors ('Weaving of biomineralization framework in rotaliid foraminifera: Implications for paleoenvironmental reconstructions', bg-2018-295) shows new insights into the pseudopodial structured during foraminiferal chamber formation, leading to better understanding of processes involved in controlling the chemical signature of the precipitated carbonate shell. By timing calcium carbonate precipitation with the structure of organic layers gives crucial information about the closeness/openness of the site of calcification, and therefore the role of passive transport (seawater exchange), which is still heavily debated. In general, the manuscript is wellstructured and well-illustrated and I just have some minor comments.

Minor comments:

- Change numbering of figures: Reference to figure 7 (page 8/ line16) before Figures 5, 6.

- 4.2 It has been suggested pores are used for gas exchange/respiration (e.g. Berthold, 1976; Leutenegger and Hansen, 1979), and their size might change with e.g. seawater oxygen level (Kuhnt et al., 2013). Would this fit with your observations? Or are the pores closed by pore plates and have no possibility to exchange?

- 4.3 Are there observed vesicles associated to seawater vacuoles? Was it possible to perform SEM-EDS on vesicles observed during chamber formation to potentially observe (amorphous phase of) calcium carbonate? Did you observe a difference in the intensity/size of vesicles during different phase of calcification and/or chamber size?

- 4.4 Implications for element distribution: When looking at element distribution across the chamber wall, it has been shown for several elements (e.g. S, Na, Mg) there is a higher concentration band near the POS. The presence of gaps in the organic layers at the initial phase of calcification compared to its absence during llater phases does explain the difference observed in element distribution (i.e. band and no-band). However, when taking Mg as an example, these Mg/Ca bands close to the POS are still much lower than expected from inorganic precipitation experiments. Based on your observations, is this because the system is not fully open, or simply because inorganic partitioning is different from foraminiferal partitioning, due to presence of other ions (inhibitors) or organic layers (adsorption)? Would this suggest that comparing foraminiferal element partitioning to inorganic precipitation experiments is not useful, since the systems are so different (organics, open/closed system etc.)?

Textual suggestions (page number/line number):

2/14 ..from seawater, which implies active ion exchange.

[Figure]

2/24 ..(Haynes, 1981), and each species..

2/25 ..modern days, during which they have..

2/29 Moreover, the tests are..

3/5 Even though the test morphology and chemical composition depend to a certain extent on the environment (), the calcification process..

5/5 For specimens fixed at different time slides during the chamber formation process..

5/15 ..the chamber formation process of A. beccarii with DIC for 59 times in total..

6/19 ..the pseudopodial activity significantly differed. A fan-shaped..

7/13 formed chamber, leaving an empty space in the new chamber..

9/16 ..corresponding to the IOL, the POS, and the OOL respectively from inner to outer side..

10/1 ..has been speculated in previous studies,..

10/6 – in other words the organic layer is part of cytoplasm.

---

## Referee Comment (RC2) · Anonymous Referee #2 · 16 Aug 2018

A paper by Nagai et al. presents novel and detailed observations on pseudopodial activities and structures during chamber formation in calcareous forams. They also verified that organic layers are a part of pseudopodia/cytoplasm. In addition, the authors also present the important finding for pore formations on calcareous walls, which have not yet been understood in detail. I think highly of their works which made progress to understand the biomineralization processes in foraminifers. I am sure that their findings would provide many ideas and hints with paleoceanographers and biogeochemists using foraminifers as tools.

I think the manuscript is generally well-structured and well-written. However, I feel the

[Figure]

Results section and Figures are still not easy to understand for most readers. The authors should take more careful about descriptions of observations, figures and their explanations. I also prefer more explanations in the figure caption to understand without reading the relevant text. I also suggest adding more close-up photographs and movie in particular for Figs. 1&2 as supplementary materials.

I assess some unsolved questions on foram biomineralization can be solved by this paper. However, some new questions are arising after reading this paper. For examples,

1) Are three organic layers produced by separate/independent pseudopodia? I think the initial stage is very important to understand this process. I suggest that the authors should present SEM photographs prior to starting the initial stage as well.

2) Thickness between OOL and POS remain the same or increasing throughout this process? In L256-257, the authors noted chamber thickening.

3) Why is the space between IOL and POS narrower than that between OOL and POS?

4) How can pore plates on POS and pore funnels on OOL align at the same locations if these are separately formed?

5) Are pore plates on POS biconvex on both side?

6) Vesicles are usually included in foraminiferal cytoplasm/pseudopodia. Why are these vesicles independently found on the surface of the OOL and POS? Were vesicles originally contained inside the OOL/POS?

7) Concerning semi-closeness/closeness at site of calcification during early/late stages, maybe your finding is related to passive/active ion transport to change Mg/Ca, but it is still speculative unless the authors verify changes in passive and active ion transport at different stages. I wonder the authors are missing the importance of the space between IOL and POS, which maybe more closed and has not yet fully understood even in this paper. Is there any possibility that differences in Mg/Ca corresponding to an IOL-POS space and a POS-OOL space? To solve this question, I suggest

showing Mg/Ca distribution map with OOL/POS lines in Fig. 7. I also wonder if elemental compositions in vesicle have any clues to solve this question.

I anticipate that some questions can be answered in the revised manuscript, but for others I look forward to future works by the authors.

Specific and technical comments

L1: Weaving? implications for paleoenvironmental reconstructions? I suggest the title rewrite as "Functional role of pseudopodia on biomineralization in rotaliid foraminifers: Implications for paleoceanographic proxies."

L22-24: I suggest rewriting as "Elemental and/or isotopic signatures of calcareous tests of Foraminifera are commonly used to reconstruct paleoenvironmental conditions."

L25: differ greatly between taxa/species/individuals/inter-chambers/intra-chambers/layers?

L26: proportional contributions from . . . > relative contributions between . . .

L27: still investigated > still under investigation/unknown/not clear

L30: Better to specify what you found for the first time

L33: POS should be explained when first mentioned in the abstract

L40: I do not think so unless the authors verify changes in passive and active ion transport at early and later stages, respectively.

L41: The "vital effect" has broad meanings. Better to specify. You may mean differences in elemental and/or isotopic ratios along chamber walls.

L42-44: Better to conclude how your findings are helpful to interpret and calibrate paleoceanographic proxies and biogeochemical cycles.

L47, Keywords: should have more important words.

[Figure]

L52-66, the first paragraph: this paragraph is jumbling about rotallids and forams in general, most of which are not directly related to the topic of this manuscript. I guess most BG readers know about forams. So better to start from the introduction of biomineralization of forams.

L71-73: Better to set this sentence as a topic sentence

L75: by experiment > by culturing experiments

L84-87: I think in situ observations and culturing experiments of foraminifera have a long history and many researches, as described in the next paragraph. I suggest deleting these sentences.

L91-95: too long noun, better to rewrite as "Superfine structure observation by . . . have been reported in order to . . ."

L100-108: The authors should more justify to use the general term "pseudopodium/a" because foraminiferal pseudopodia are usually named as granuloreticulopodia. I would only agree with the authors if foraminifers do not produce any dynamic net-like structures with no any granules visible during chamber formation.

L116-117: The "POS" used to be called as POM (Primary Organic Membrane) in Hemleben et al. (1986).

L116-118: OOL and IOL are not first mentioned

L124: The new term "organic scaffolding" are not easy to imagine and not mentioned as an important term throughout the manuscript. I suggest the author redefine the term "anlage" to confine organic layers.

L126-127: Use POS, OOL and IOL consistently throughout the text except for first mentioned.

L130: natural state?

L130-131: electron microscopy>SEM/TEM?

L132, (SOC): Move to L129 that is first mentioned

L150-151: paleoenvironments>palaeoceanographic proxies; predicting responses to ongoing climate change > how?

L153: Better to rewrite as "within a hyaline calcareous wall using the benthic foraminifera"

L157, SEM: Define when first mentioned.

L167: De Nooijer et al., 2009? Check all years of references in the text. I found some typos in other refs as well.

L208: The first paragraph of the Results section is just an outline and unnecessary. Delete or partly move to the method section.

L223: the last existing calcified chamber > the last chamber

L224: characteristic > morphology

L226: delete "from then"

L231, an aggregation of cytoplasm: Indicate where and which part in the figure,

L233: retracts until where?

L238-239: fine and short pseudopodia? I cannot see it. Need more close-up photos.

L240-241: A brighter band of particles? I cannot see it.

L242: beyond? inside?

L243: smooth? Fig. 1C looks smoother than 1D

L251, Calcium carbonate: How do you know it?

L252: I think the overall outline and size are fixed at earlier stages (the middle stage).

L253: Hard to see pseudopodial movements. Do you have a movie?

L253: Open triangles in Fig. 2A?

L256-257: How do you know the chamber wall getting thicker?

L262: The usual type of pseudopodia movement means reticulopodia?

L266-267: Move to the method section

L266-277: This paragraph with Fig. 3 should move to the Discussion section because Fig. 3 are mostly schematic models and your interpretations based on observations.

L274: gray in Figures 2-4?

L274: um?

L292, vesicle: How do you identify it? Vesicles are usually included in foraminiferal cytoplasm/pseudopodia. Why are these vesicles independently found on the surface of OOL and POS?

L295, pseudopodia: how do you identify it?

L311, needle-like structure: Show in the figure.

L316-317: Show in the figure

L328-329: Which are algal cysts in Fig. 5A?

L335: period>stage

L353: OOL had toward the inner side?

L376-377: Indicate which photos clearly show this.

L387: Cader>Cadre, Ni>Ní based on references

Figs. 1&2: Add color legend in the figure; indicate initial, middle, late stages in the figure; hard to see any bubbles and pseudopodia inside chambers; in Fig. 1C, the

frame of new chamber are magenta?; I prefer more explanations in the caption to understand without reading the relevant text; open triangle?; Did you identify calcareous wall by polarized microscope?: any more magnified images? You should have used a fluorescent dye to observe cytoplasm more clearly.

Fig. 3: move after Figs. 4-7; indicate initial, middle, late stages in the figure; for A, indicate which part of close-up in B; colors of outer (blue) and inner sides (purple) are confusing with vesicle and pseudopodia.; the shape of carbonate crystals looks like needles. is it OK?; What are the purple colored polygonal shape on the POS?

Figs. 4-6: Indicate differences between dotted lines and thick lines.

Fig. 7: Indicate OOL and IOL lines; Add Mg signal and Mg/Ca data

---

## Author Comment (AC1) · 31 Aug 2018

Dear Inge,

We are pleased to submit the revised version of our Research Article manuscript "Weaving of biomineralization framework in rotaliid foraminifera: Implications for paleoenvironmental reconstructions" (bg-2018-295). We appreciated your constructive criticisms and comments, and we thank you for providing this opportunity for us to improve this manuscript and submit a revised version.

A point-by-point response to comments is included below. All files for the revised

manuscript (tracked changes and clean versions, figures, table, supplementary files) are contained in the .ZIP file uploaded with this revision.

We hope the present version is acceptable for publication in Biogeosciences.

Best regards,

Yukiko Nagai JAMSTEC
* * *
The study by Nagai and co autors ('Weaving of biomineralization framework in rotaliid foraminifera: Implications for paleoenvironmental reconstructions', bg-2018-295) shows new insights into the pseudopodial structured during foraminiferal chamber formation, leading to better understanding of processes involved in controlling the chemical signature of the precipitated carbonate shell. By timing calcium carbonate precipitation with the structure of organic layers gives crucial information about the closeness/openness of the site of calcification, and therefore the role of passive transport (seawater exchange), which is still heavily debated. In general, the manuscript is well-structured and well-illustrated and I just have some minor comments. We are grateful for these encouraging words and your constructive review of our manuscript.

Minor comments:

- Change numbering of figures: Reference to figure 7 (page 8/ line16) before Figures 5, 6. We preferred to keep the SEM plates of the microstructures together. We have thus added the following to the beginning of section 3.2 to justify the numbering (we also moved Fig. 3, the schematics, to the end as the new Fig. 7 according to a comment from Reviewer 2). 'Figures 3-6 show microstructures of different stages of chamber formation seen by the SEM.

- 4.2 It has been suggested pores are used for gas exchange/respiration (e.g. Berthold, 1976; Leutenegger and Hansen, 1979), and their size might change with e.g. seawater oxygen level (Kuhnt et al., 2013). Would this fit with your observations? Or are

the pores closed by pore plates and have no possibility to exchange? As O2 and CO2 used in respiration are nonpolar molecules, they are able to pass through the cell membrane. As such, the existence of pore plates made from cytoplasm seen in the present study should not influence respiration. Whether their size change with environment differences is a subject for future studies. Added the following in Discussion: 'Pores have been suggested to be used for respiration (e.g., Berthold, 1976; Leutenegger & Hansen, 1979). As O2 and CO2 used in respiration are nonpolar molecules, they are able to pass through the cell membrane. As such, the existence of pore plates made from cytoplasm seen in the present study should not influence respiration.'

- 4.3 Are there observed vesicles associated to seawater vacuoles? Was it possible to perform SEM-EDS on vesicles observed during chamber formation to potentially observe (amorphous phase of) calcium carbonate? Did you observe a difference in the intensity/size of vesicles during different phase of calcification and/or chamber size? In the present study, we were unable to observe amorphous phase of calcium carbonate within the 'vesicles'. In fact, as another reviewer pointed out, we do not have sufficient evidence to prove that these spherical structures indeed correspond to vesicles (and they are outside the cell!). Therefore, we have changed the terminology from 'vesicle' to 'spherical structure' throughout the revised MS. Although we did find these spherical structures with high Ca signals in our SEM-EDS analyses, we do not know for sure whether these signals are indeed from within the structures or from tests underneath. Making this clear requires future elemental analyses using thin-sliced embedded material.

- 4.4 Implications for element distribution: When looking at element distribution across the chamber wall, it has been shown for several elements (e.g. S, Na, Mg) there is a higher concentration band near the POS. The presence of gaps in the organic layers at the initial phase of calcification compared to its absence during llater phases does explain the difference observed in element distribution (i.e. band and no-band). Yes, we agree. However, when taking Mg as an example, these Mg/Ca bands close to the POS

are still much lower than expected from inorganic precipitation experiments. Based on your observations, is this because the system is not fully open, or simply because inorganic partitioning is different from foraminiferal partitioning, due to presence of other ions (inhibitors) or organic layers (adsorption)? Unfortunately, from the present results it is not possible to say how much seawater actually passes through the gaps in organic layers during chamber formation. From the beginning, the Mg content of the fluid is already decreased even the system is not fully closed, because the observed gaps are not sufficiently large to exchange seawater between the site of calcification and the exterior. But this is just speculative at this point. Would this suggest that comparing foraminiferal element partitioning to inorganic precipitation experiments is not useful, since the systems are so different (organics, open/closed system etc.)? We consider comparing foraminiferal elemental partitioning to inorganic precipitation still meaningful, as the partitioning from fluid to crystal follows the same chemical laws in both organic and inorganic systems. The key difference is that the elemental contents of the fluid in the site of calcification is strongly controlled in biomineralisation processes (and very different from inorganic processes), and in the future research we hope to clarify the elemental composition of the fluid at the site of calcification. We have added the following to Discussion: 'The elemental partitioning in foraminiferal tests must be strongly controlled through the elemental composition of the fluid in the SOC, which is a key subject for future studies.'

Textual suggestions (page number/line number): 2/14 ..from seawater, which implies active ion exchange. Changed as suggested.

2/24 ..(Haynes, 1981), and each species.. Changed as suggested.

2/25 ..modern days, during which they have.. Changed as suggested.

2/29 Moreover, the tests are.. Changed as suggested.

3/5 Even though the test morphology and chemical composition depend to a certain extent on the environment (), the calcification process.. Changed as suggested.

5/5 For specimens fixed at different time slides during the chamber formation process.. Changed as suggested.

5/15 ..the chamber formation process of A. beccarii with DIC for 59 times in total.. Changed as suggested.

6/19 ..the pseudopodial activity significantly differed. A fan-shaped.. Changed as suggested.

7/13 formed chamber, leaving an empty space in the new chamber.. Changed as suggested.

9/16 ..corresponding to the IOL, the POS, and the OOL respectively from inner to outer side.. Changed as suggested.

10/1 ..has been speculated in previous studies,.. Changed as suggested.

10/6 – in other words the organic layer is part of cytoplasm. Changed as suggested.

Please also note the supplement to this comment:
https://www.biogeosciences-discuss.net/bg-2018-295/bg-2018-295-AC1-supplement.zip

---

## Author Comment (AC3) · 31 Aug 2018

Dear Hiroshi,

We are pleased to submit the revised version of our Research Article manuscript "Weaving of biomineralization framework in rotaliid foraminifera: Implications for paleoenvironmental reconstructions" (bg-2018-295), which received two reviews for Biogeosciences.

We appreciated the constructive criticisms and comments from the two reviewers, and we thank you for providing this opportunity for us to improve this manuscript and submit

a revised version.

A point-by-point response to comments and all manuscript files are uploaded in the .ZIP files under responses to each reviewer comments.

We hope the present version is acceptable for publication in Biogeosciences.

Best regards,

Yukiko Nagai JAMSTEC

---

## Author Response (AR1)

Dear Professor Kitazato,

We are pleased to submit a further revised version of our Research Article manuscript "**Weaving of biomineralization framework in rotaliid foraminifera: Implications for paleoenvironmental reconstructions**" (bg-2018-295), which received two reviews for *Biogeosciences*.

We appreciated the constructive criticisms and comments from the two reviewers, and we thank you for providing this opportunity for us to improve this manuscript and submit a revised version. Furthermore, in response to the suggestion from Professor Kitazato, we added more detailed discussions along the reviewers' comments, in order to answer and discuss openly the topics raised.

A point-by-point response to comments is included below.

We hope the present version is acceptable for publication in *Biogeosciences*.

Best regards,

Yukiko Nagai

**RESPONSE TO REVIEWER 1 (Inge van Dijk)**

**The study by Nagai and co autors ('Weaving of biomineralization framework in rotaliid foraminifera: Implications for paleoenvironmental reconstructions', bg-2018-295) shows new insights into the pseudopodial structured during foraminiferal chamber formation, leading to better understanding of processes involved in controlling the chemical signature of the precipitated carbonate shell. By timing calcium carbonate precipitation with the structure of organic layers gives crucial information about the closeness/openness of the site of calcification, and therefore the role of passive transport (seawater exchange), which is still heavily debated. In general, the manuscript is well-structured and well-illustrated**

**and I just have some minor comments.**

> We are grateful for these encouraging words and your constructive review of our manuscript.

**Minor comments:**

**- Change numbering of figures: Reference to figure 7 (page 8/ line16) before Figures 5, 6.**

> We preferred to keep the SEM plates of the microstructures together. We have thus added the following to the beginning of section 3.2 to justify the numbering (we also moved Fig. 3, the schematics, to the end as the new Fig. 7 according to a comment from Reviewer 2).
>
> 'Figures 3-6 show microstructures of different stages of chamber formation seen by the SEM.

**- 4.2 It has been suggested pores are used for gas exchange/respiration (e.g. Berthold, 1976; Leutenegger and Hansen, 1979), and their size might change with e.g. seawater oxygen level (Kuhnt et al., 2013). Would this fit with your observations? Or are the pores closed by pore plates and have no possibility to exchange?**

> As $O_2$ and $CO_2$ used in respiration are nonpolar molecules, they are able to pass through the cell membrane. As such, the existence of pore plates made from cytoplasm seen in the present study should not influence respiration. Whether their size change with environment differences is a subject for future studies. Added the following in Discussion:
>
> 'Pores have been suggested to be used for respiration (e.g., Berthold, 1976; Leutenegger & Hansen, 1979). As $O_2$ and $CO_2$ used in respiration are nonpolar molecules, they are able to pass through the cell membrane. As such, the existence of pore plates made from cytoplasm seen in the present study should not influence respiration.'

**- 4.3 Are there observed vesicles associated to seawater vacuoles? Was it possible to perform SEM-EDS on vesicles observed during chamber formation to potentially observe (amorphous phase of) calcium carbonate? Did you observe a difference in the intensity/size of vesicles during different phase of calcification and/or chamber size?**

In the present study, we were unable to observe amorphous phase of calcium carbonate within the 'vesicles'. In fact, as another reviewer pointed out, we do not have sufficient evidence to prove that these spherical structures indeed correspond to vesicles (and they are outside the cell!). Therefore, we have changed the terminology from 'vesicle' to 'spherical structure' throughout the revised MS. Although we did find these spherical structures with high Ca signals in our SEM-EDS analyses (Figure 6B), we could not know for sure whether these signals are indeed from within the structures or from tests underneath by limitation of the current measurement technology. Added the following in Discussion:

'The number of spherical structures increased as the chamber formation progressed. There is a variation in the size of spherical structure from ca. 50 nm - 1 μm. There are relatively more small ones at the Initial stage, and relatively more large ones at the Late stage, but all sizes of the spherical structures are found across all stages.'

**- 4.4 Implications for element distribution: When looking at element distribution across the chamber wall, it has been shown for several elements (e.g. S, Na, Mg) there is a higher concentration band near the POS. The presence of gaps in the organic layers at the initial phase of calcification compared to its absence during llater phases does explain the difference observed in element distribution (i.e. band and no-band).**

Yes, we agree.

**However, when taking Mg as an example, these Mg/Ca bands close to the POS are still much lower than expected from inorganic precipitation experiments. Based on your observations, is this because the system is not fully open, or simply because inorganic partitioning is different from foraminiferal partitioning, due to presence of other ions (inhibitors) or organic layers (adsorption)? Would this suggest that comparing foraminiferal element partitioning to inorganic precipitation experiments is not useful, since the systems are so different (organics, open/closed system etc.)?**

From the present results it is not possible to say how much seawater actually passes through the gaps in organic layers during chamber formation. From the beginning, the Mg content of the fluid is already decreased even the system is not fully closed, because the observed gaps are not sufficiently large to exchange seawater between the site of calcification and the exterior. But this is

just speculative at this point.

We consider comparing foraminiferal elemental partitioning to inorganic precipitation still meaningful, as the partitioning from fluid to crystal follows the same chemical laws in both organic and inorganic systems. The key difference is that the elemental contents of the fluid in the site of calcification is strongly controlled in biomineralisation processes (and very different from inorganic processes). The elemental analysis of fluid at the site of calcification, however, is still currently unmeasurable due to technical limitations. Nevertheless, because magnesium ion is an inhibitor of calcification, it can be speculated that during biomineralization magnesium ions are actively discriminated and removed from the fluid at the site of calcification. Therefore, as pointed out by the reviewers, it is presumed that calcite with low Mg / Ca precipitates around the POS.

We have added the following to Discussion:

"The elemental analysis of fluid at the site of calcification, however, is still currently unmeasurable due to technical limitations. Nevertheless, because magnesium ion is an inhibitor of calcification, it can be speculated that during biomineralization magnesium ions are actively discriminated and removed from the fluid at the site of calcification (Zeebe and Sanyal, 2002). Therefore, it is presumed that calcite with low Mg/Ca precipitates even around the POS. It is reported in many species that the foraminiferal Mg/Ca is high around the POS, but it is still much lower than Mg/Ca estimated from inorganic precipitation experiments (de Nooijer et al., 2014). The elemental partitioning in foraminiferal tests must be strongly controlled through the elemental composition of the fluid in the SOC, which is a key subject for future studies."

**Textual suggestions (page number/line number):**
**2/14 ..from seawater, which implies active ion exchange.**
Changed as suggested.

**2/24 ..(Haynes, 1981), and each species..**
Changed as suggested.

**2/25 ..modern days, during which they have..**
Changed as suggested.

**2/29 Moreover, the tests are..**

Changed as suggested.

**3/5 Even though the test morphology and chemical composition depend to a certain extent on the environment (), the calcification process..**

Changed as suggested.

**5/5 For specimens fixed at different time slides during the chamber formation process..**

Changed as suggested.

**5/15 ..the chamber formation process of A. beccarii with DIC for 59 times in total..**

Changed as suggested.

**6/19 ..the pseudopodial activity significantly differed. A fan-shaped..**

Changed as suggested.

**7/13 formed chamber, leaving an empty space in the new chamber..**

Changed as suggested.

**9/16 ..corresponding to the IOL, the POS, and the OOL respectively from inner to outer side..**

Changed as suggested.

**10/1 ..has been speculated in previous studies,..**

Changed as suggested.

**10/6 – in other words the organic layer is part of cytoplasm.**

Changed as suggested.

**RESPONSE TO REVIEWER 2**

**A paper by Nagai et al. presents novel and detailed observations on pseudopodial activities and structures during chamber formation in calcareous forams. They also verified that organic layers are a part of pseudopodia/cytoplasm. In addition,**

**the authors also present the important finding for pore formations on calcareous walls, which have not yet been understood in detail. I think highly of their works which made progress to understand the biomineralization processes in foraminifers. I am sure that their findings would provide many ideas and hints with paleoceanographers and biogeochemists using foraminifers as tools.**

> Many thanks for your positive comments and we appreciated your very thorough review of our manuscript.

**I think the manuscript is generally well-structured and well-written. However, I feel the Results section and Figures are still not easy to understand for most readers. The authors should take more careful about descriptions of observations, figures and their explanations. I also prefer more explanations in the figure caption to understand without reading the relevant text. I also suggest adding more close-up photographs and movie in particular for Figs. 1&2 as supplementary materials.**

> We can agree with these comments in general, and we have increased textual explanations and details on figure captions according to the comments (which are detailed below). We also added higher magnification photos and a video as supplemental materials. As reviewer 2 pointed out, the distribution of pseudopodia during chamber formation is much better visible with a video We add supplementary movie and mentioned about the movie in the text of result section.

**I assess some unsolved questions on foram biomineralization can be solved by this paper. However, some new questions are arising after reading this paper. For examples,**
**1) Are three organic layers produced by separate/independent pseudopodia? I think the initial stage is very important to understand this process. I suggest that the authors should present SEM photographs prior to starting the initial stage as well.**

> We consider that all three organic sheets are formed with branched pseudopodia extending from the aperture. We expect these pseudopodia themselves are ultimately expanded from single root, but separate branches are forming each organic sheet. Our recent study (Nagai et al., 2018) showed that all sheets actually converge at the pore plate. Further, pore plates and pore funnels smoothly peeled off each other, which suggests that each sheet is

formed from independent pseudopodia. We think the plates and funnels are gently adhered to each other. We must agree that the very initial stage of these sheet construction must be important, as can be inferred from the details of organic sheet arrangement clarified in the present study.

Regarding early stages before the Initial stage, it was already shown in our recent study (Nagai et al., 2018) that the three sheets (OOL, POS, IOL) are separate in the very early stage (although ultimately from a single root), and thus we refer to SEM figures contained within that study.

Added the following to Discussion to make this clear:

'According to a recent study (Nagai et al., 2018), the three sheets (OOL, POS and IOL) appears initially to be independent even at the very early stage of chamber formation during the total thickness of the whole organic sheet being less than 1 μm. We expect these organic sheets themselves are ultimately expanded from single root, but separate branches of pseudopodia are forming each organic sheet.'

**2) Thickness between OOL and POS remain the same or increasing throughout this process? In L256-257, the authors noted chamber thickening.**

The distance between OOL and POS increased along the growth of the calcareous wall. Added: 'the chamber wall of which thickens (overall distance between OOL and POS increased over time).

**3) Why is the space between IOL and POS narrower than that between OOL and POS?**

This is caused by the difference of growth rate of calcareous naterual between the inner side and the outer side. Assuming that the materials (Ca2+, Carbon, among others) are transported from the seawater, it can be presumed that the inner side will become thinner because the chamber wall is formed and material transportation is more restricted in the inner side. Added the following to Discussion:

'The reason why the space between IOL and POS is narrower than between OOL and POS (meaning the inner calcareous layer is thinner than the outer) is presumably caused by the difference of growth rate of calcareous material

between the inner side and the outer side. Assuming that the materials for chamber formation are transported from the seawater, it can be presumed that the inner side will become thinner because the chamber wall is formed and material transportation is more restricted in the inner side.'

**4) How can pore plates on POS and pore funnels on OOL align at the same locations if these are separately formed?**

At the beginning of construction, all three sheets (OOL, POS and IOL) are converged at the pore site (as was shown by Nagai et al. 2018), in which case these are not truly separately formed and explains the alignment of pore plates and funnels (i.e., we interpret that the sites of pore plates and funnels formation is aligned in very early stage when the sheets are still converged at the pore site). Added the following to Discussion:

'Pore plates and pore funnels smoothly peeled off from one another (Fig. 3B and 7A), suggesting that pore plates and pore funnels belong to independent organic sheets formed from separated pseudopodia. Pore plates and funnels were gently adhered to each other before calcification started. It is thought that pore plates and pore funnels are formed simultaneously face to face, during the organic sheet formation. Pore plates and pore funnels likely function as anchors that hold together all three sheets to result ultimately in a smooth calcareous wall.'

**5) Are pore plates on POS biconvex on both side?**

Pore plates seems to be dented at the IOL side (see Fig. 3C in Nagai et al. 2018). Added the following in Results:

'Pore plates seems to be dented at the IOL side, from a previous study (see Fig. 3C in Nagai et al. 2018).'

**6) Vesicles are usually included in foraminiferal cytoplasm/pseudopodia. Why are these vesicles independently found on the surface of the OOL and POS? Were vesicles originally contained inside the OOL/POS?**

Indeed, we agree that vesicles are usually found inside the cell, as the reviewer mentioned. There is no sufficient evidence in the present study to prove that these are vesicles, and thus we changed 'vesicles' to 'spherical structures' across the entire manuscript.

**7) Concerning semi-closeness/closeness at site of calcification during early/late stages, maybe your finding is related to passive/active ion transport to change Mg/Ca, but it is still speculative unless the authors verify changes in passive and active ion transport at different stages. I wonder the authors are missing the importance of the space between IOL and POS, which maybe more closed and has not yet fully understood even in this paper. Is there any possibility that differences in Mg/Ca corresponding to an IOL-POS space and a POS-OOL space? To solve this question, I suggest showing Mg/Ca distribution map with OOL/POS lines in Fig. 7. I also wonder if elemental compositions in vesicle have any clues to solve this question.**

We approve this point by the reviewer, and agree that the actual amount of seawater exchanged is a key subject to explore in the future. Regarding the differences in Mg/Ca between IOL-POS vs POS-OOL space, measuring the elemental distribution in such a narrow gap is very difficult with existing techniques. This is because chemical compositions are not well preserved during the conventional process of sample prep for electron microscopy. We carried out EDS analyses with the same interest in mind, but have not achieved sufficient results for a publication because Mg signals of the early deposited calcite was below detection level. However, we agree that this is a very important point to mention, and included the following in Discussion:

"The elemental composition of the inner calcified layer formed between the IOL and the POS is probably more closed and strongly affected by cellular processes than the outer calcified layer between the OOL and the POS. Therefore, the magnesium contents of the inner layer may differ from the outer layer with pure calcite may be precipitated at the inner side."

Instead of that, we discussed about elemental (Mg/Ca) heterogeneity within a single wall along the chamber formation process with closeness of the site of calcification, as Reviewer#1 pointed out:

" The elemental analysis of fluid at the site of calcification, however, is still currently unmeasurable due to technical limitations. Nevertheless, because magnesium ion is an inhibitor of calcification, it can be speculated that during biomineralization magnesium ions are actively discriminated and removed from the fluid at the site of calcification (Zeebe and Sanyal, 2002). Therefore,

it is presumed that calcite with low Mg/Ca precipitates even around the POS. It is reported in many species that the foraminiferal Mg/Ca is high around the POS, but it is still much lower than Mg/Ca estimated from inorganic precipitation experiments (de Nooijer et al., 2014). The elemental partitioning in foraminiferal tests must be strongly controlled through the elemental composition of the fluid in the SOC, which is a key subject for future studies."

**I anticipate that some questions can be answered in the revised manuscript, but for others I look forward to future works by the authors.**

Many thanks. We have attempted to do so, as detailed above.

**Specific and technical comments**

**L1: Weaving? implications for paleoenvironmental reconstructions? I suggest the title rewrite as "Functional role of pseudopodia on biomineralization in rotaliid foraminifers: Implications for paleoceanographic proxies."**

We can agree in part, in that saying 'reconstructions' is going too far. So we changed the second part of the title as suggested. It now reads: '*Weaving of biomineralization framework in rotaliid foraminifera: Implications for paleoceanographic proxies*". We hope this is ok.

**L22-24: I suggest rewriting as "Elemental and/or isotopic signatures of calcareous tests of Foraminifera are commonly used to reconstruct paleoenvironmental conditions."**

Changed as suggested.

**L25: differ greatly between taxa/species/individuals/inter-chambers/ intrachambers/layers?**

Added '…, as well as between taxa, species, individuals, etc.'

**L26: proportional contributions from : : : > relative contributions between : : :**

Corrected.

**L27: still investigated > still under investigation/unknown/not clear**

Changed to '… still under investigation'.

**L30: Better to specify what you found for the first time**

To make it clear we moved 'for the first time' to the end, it now reads: 'We document triple organic layers sandwiching carbonate precipitation sites for the first time'.

**L33: POS should be explained when first mentioned in the abstract**

Agreed and changed to 'primary organic sheet'.

**L40: I do not think so unless the authors verify changes in passive and active ion transport at early and later stages, respectively.**

We agree with you and therefore changed the expression to make it more mild: 'provides insight towards resolving' instead of 'resolved'.

**L41: The "vital effect" has broad meanings. Better to specify. You may mean differences in elemental and/or isotopic ratios along chamber walls.**

Specified as follows: 'The 'vital effect', specifically elemental and isotopic ratios along chamber walls,...'

**L42-44: Better to conclude how your findings are helpful to interpret and calibrate paleoceanographic proxies and biogeochemical cycles.**

To address this we reorganised the last few sentences of the introduction, as follows, which we hope helps:

'Our study provides insight towards resolving a key 'missing piece' in understanding foraminiferal calcification though culture experiments and in-depth observations of living animals. Our findings contribute to interpreting and understanding biogeochemical proxies by showing that the 'vital effect', specifically elemental and isotopic ratios along chamber walls, is directly linked to spatio-temporal organization of the 'biomineralization sandwich' controlled by the three major organic layers.

**L47, Keywords: should have more important words.**

*Biogeosciences* does not actually require keywords, so now they have been removed.

**L52-66, the first paragraph: this paragraph is jumbling about rotallids and forams**

**in general, most of which are not directly related to the topic of this manuscript. I guess most BG readers know about forams. So better to start from the introduction of biomineralization of forams.**

We can agree with this and have deleted the first paragraph.

**L71-73: Better to set this sentence as a topic sentence**

Done.

**L75: by experiment > by culturing experiments**

Corrected as suggested.

**L84-87: I think in situ observations and culturing experiments of foraminifera have a long history and many researches, as described in the next paragraph. I suggest deleting these sentences.**

Deleted as suggested.

**L91-95: too long noun, better to rewrite as "Superfine structure observation by : : : have been reported in order to : : :"**

Modified as suggested.

**L100-108: The authors should more justify to use the general term "pseudopodium/a" because foraminiferal pseudopodia are usually named as granuloreticulopodia. I would only agree with the authors if foraminifers do not produce any dynamic net-like structures with no any granules visible during chamber formation.**

Your suggestion is true, but we prefer to use pseudopodia as it is a more general term. We modified the sentence as follows to make this clear:

'Foraminiferal pseudopodia are usually named granuloreticulopodia (see Travis and Bowser, 1991) to define a granular reticulated pseudopodium responsible for feeding, digestion and locomotion; in the present paper we will simply use pseudopodia as it is a more general term.'

**L116-117: The "POS" used to be called as POM (Primary Organic Membrane) in Hemleben et al. (1986).**

Changed to '…the one in the middle was initially named the 'Primary organic membrane' (POM) (Hemleben et al., 1986) but later changed to 'Primary

organic sheet' (POS) (Erez, 2003)'

**L116-118: OOL and IOL are not first mentioned**
> True. Changed to simply OOL and IOL instead of spelling out the whole name.

**L124: The new term "organic scaffolding" are not easy to imagine and not mentioned as an important term throughout the manuscript. I suggest the author redefine the term "anlage" to confine organic layers.**
> Since *Anlage* has been defined differently by different people, we disagree and would like to refresh with a new term to avoid confusion in the future.

**L126-127: Use POS, OOL and IOL consistently throughout the text except for first mentioned.**
> Changed to simply OOL and IOL instead of spelling out the whole name.

**L130: natural state?**
> Changed to 'well-preserved morphology'.

**L130-131: electron microscopy>SEM/TEM?**
> Changed as suggested.

**L132, (SOC): Move to L129 that is first mentioned**
> Changed as suggested.

**L150-151: paleoenvironments>palaeoceanographic proxies**
> Changed as suggested

**; predicting responses to ongoing climate change > how?**
> Deleted this part of the sentence.

**L153: Better to rewrite as "within a hyaline calcareous wall using the benthic foraminifera"**
> Changed as suggested.

**L157, SEM: Define when first mentioned.**
> This was defined already in the introduction.

**L167: De Nooijer et al., 2009? Check all years of references in the text. I found some typos in other refs as well.**

      Corrected and checked all years of references.

**L208: The first paragraph of the Results section is just an outline and unnecessary. Delete or partly move to the method section.**

      Deleted.

**L223: the last existing calcified chamber > the last chamber**

      Modified as suggested.

**L224: characteristic > morphology**

      Modified as suggested.

**L226: delete "from then"**

      Deleted.

**L231, an aggregation of cytoplasm: Indicate where and which part in the figure,**

      Added reference to Figure 1B.

**L233: retracts until where?**

      Added 'until the surface of the newly forming chamber'.

**L238-239: fine and short pseudopodia? I cannot see it. Need more close-up photos.**

      We added an enlarged part to Figure 1C, we hope this helps.

**L240-241: A brighter band of particles? I cannot see it.**

      Changed to simply saying 'bright band'.

**L242: beyond? inside?**

      Changed to 'inside' as suggested.

**L243: smooth? Fig. 1C looks smoother than 1D**

      Deleted this part of the sentence.

**L251, Calcium carbonate: How do you know it?**

Added to 'material, inferred to be calcium carbonate,' at beginning of the paragraph.

**L252: I think the overall outline and size are fixed at earlier stages (the middle stage).**

We removed 'size' but the outline morphology is actually changing since the middle stage so we left it.

**L253: Hard to see pseudopodial movements. Do you have a movie?**

We have added a movie as Supplementary Video.

**L253: Open triangles in Fig. 2A?**

Deleted 'open triangles' (this referred to an earlier version of the figure, apologies).

**L256-257: How do you know the chamber wall getting thicker?**

This is clearly visible in the supplementary video, so we added reference to that video:

'(from Figure 2B–C; also see Supplementary Video)'

**L262: The usual type of pseudopodia movement means reticulopodia?**

Added 'usual type of pseudopodia movement (typical of reticulopodia)'.

**L266-267: Move to the method section**

We do not think this can be moved to Methods because without results from the observation of the stages, we would not have been able to divide the stages as such. So we have left this section here.

**L266-277: This paragraph with Fig. 3 should move to the Discussion section because Fig. 3 are mostly schematic models and your interpretations based on observations.**

Moved.

**L274: gray in Figures 2-4?**

Corrected.

**L274: um?**

    Corrected.

**L292, vesicle: How do you identify it? Vesicles are usually included in foraminiferal cytoplasm/pseudopodia. Why are these vesicles independently found on the surface of OOL and POS?**

    This is a good point in that we cannot clearly identify these spherical structures as vesicles with certainly, at this point. Therefore we have changed the name to 'spherical structures' throughout the manuscript and deleted the assumption that these represent vesicles.

**L295, pseudopodia: how do you identify it?**

    We can only identify pseudopodia by morphology, that the elongated structures are inferred to be pseudopodia. Added the following to make this clear: 'elongated structures, inferred to be pseudopodia,'

**L311, needle-like structure: Show in the figure.**

    Added panels to Figure 5E to show this.

**L316-317: Show in the figure**

    Added arrowheads in Figure 5E to show the gaps.

**L328-329: Which are algal cysts in Fig. 5A?**

    These overlay the OOL, so we added 'Algal cysts including *Dunaliella* individuals can be seen overlaying the OOL'.

**L335: period>stage**

    Corrected.

**L353: OOL had toward the inner side?**

    This is a part of a line we thought we have removed, apologies. Deleted now.

**L376-377: Indicate which photos clearly show this.**

    Added '(Figure 4C-E, Supplementary Video)'

**L387: Cader>Cadre, Ni>Ní based on references**

> Corrected.

**Figs. 1&2: Add color legend in the figure;**

> Added.

**indicate initial, middle, late stages in the figure;**

> Indicated

**hard to see any bubbles and pseudopodia inside chambers;**

> This is more visible in the Supplementary Video which we have added now, these represent structures within the chambers.

**in Fig. 1C, the frame of new chamber are magenta?;**

> No, in 1C there is no calcified parts yet, and magenta indicates calcified wall. So this is only present in 1D. We made this more easily visible in 1D now.

**I prefer more explanations in the caption to understand without reading the relevant text;**

> We added more comprehensive explanations to Figs 1 & 2 in the revised captions, as follows. It should be understandable without having to read the relevant text now.

> 'Figure 1: Time series observation of chamber formation by optical microscopy, as seen in the individual observed on December 7th, 2017 (see Table 1). The initial stage of chamber formation, where the organic framework is built, depicted by A-B. A: Beginning of chamber formation, defined as 0 minute from the start, indicated by a dense radiating spray of pseudopodial network. B: 9 minutes, when an aggregation of cytoplasm becomes visible around the aperture of the last existing chamber. As this cytoplasm expands, the pseudopodial network starts to retract to the surface of the new chamber to complete the framework. The middle stage, where the organic framework is prepared for calcium carbonate precipitation which begins at near the end of this stage, takes place between 15 minutes to 60 minutes, as depicted by C-D. C: 27 minutes, cytoplasm concentrates and outline of newly forming chamber wall now clearly visible, pseudopodia still just visible on the surface. D: 41 minutes, pseudopodial retracts inside the forming chamber wall. Left: optical microscopy image. Right: the same image with schematic overlay; colour legend: deep purple = pseudopodia; light purple = cytoplasm; magenta = calcium carbonate in the newly forming chamber; yellow = previously formed

chambers.

Figure 2: Time series observation of chamber formation by optical microscopy (continued), as seen in the individual observed on December 7th, 2017 (see Table 1). The late stage of chamber formation, where calcium carbonate is extensively precipitated and chamber wall is thickened, taking place from around 60 minutes after the start of chamber formation (total time varies considerably among individuals). A: 65 minutes, pseudopodia expands again to form a dense network but in thicker strands than seen in previous stages. B: 100 minutes, a network of pseudopodia is seen in the new chamber, the chamber wall of which thickens. C: 124 minutes, chamber wall thickening continues. D: 180 minutes, chamber wall thickening is nearly and the pseudopodial network begin to disappear, indicating that the end of the chamber formation process is near (actual completion was at 248 minutes for this individual). Left: optical microscopy image. Right: the same image with schematic overlay; colour legend: deep purple = pseudopodia; light purple = cytoplasm; magenta = calcium carbonate in the newly forming chamber; yellow = previously formed chambers.'

**open triangle?;**

Deleted, this was from an earlier version which should have been removed before submission.

**Did you identify calcareous wall by polarized microscope?:**

We inferred the brighter parts to be calcium carbonate based on images from differential interference contrast (DIC) microscopy.

**any more magnified images?**

We added images in original resolution as supplementary material.

**You should have used a fluorescent dye to observe cytoplasm more clearly.**

That is a possibility for future research, but we consider that the activity of pseudopodia is already clearly seen in the supplementary video taken with DIC.

**Fig. 3: move after Figs. 4-7;**

Moved.

**indicate initial, middle, late stages in the figure;**

Indicated.

**for A, indicate which part of close-up in B;**

B is not a close-up of A but instead is a cross-section.

**colors of outer (blue) and inner sides (purple) are confusing with vesicle and pseudopodia.;**

Background colours are deleted.

**the shape of carbonate crystals looks like needles. is it OK?;**

Yes. We have added a reference to Figure 4E, newly added to show needle-like crystals.

**What are the purple colored polygonal shape on the POS?**

These are holes, we removed the colours.

**Figs. 4-6: Indicate differences between dotted lines and thick lines.**

Added 'Thick lines indicate membranous pseudopodia and dotted lines indicate framework pseudopodia.'

**Fig. 7: Indicate OOL and IOL lines;**

Added this on Panel C.

**Add Mg signal and Mg/Ca data**

The Mg signals were too weak, therefore we left it out.

[revised manuscript text omitted]

indicating they probably ultimately expand from a single root. This, however, warrants confirmation by higher resolution investigation of the very start of the initial stage in future studies.

The importance of organic layers in the early stages of chamber formation has been speculated in the previous studies, but little was known about its origin. It was previously thought that the organic layer was secreted from the pseudopodia (e.g., Angell, 1967; Röttiger, 1974; Hemleben *et al*., 1986), and Spindler and Röttiger (1973) reported that the organic layer seems to be connected with pseudopodia. These studies were largely limited in that their magnification (only light microscopy was available then) was not sufficient resolution to observe the detailed process. The process documented herein provides evidence for an entirely novel model in that the pseudopodia itself weaves the organic layers (Figure 3C-E, Supplementary Video)– in other words the organic layer is the part of cytoplasm.

The reason why the space between IOL and POS is narrower than between OOL and POS (meaning the inner calcareous layer is thinner than the outer) is presumably caused by the difference of growth rate of calcareous material between the inner side and the outer side. Assuming that the materials for chamber formation are transported from the seawater, it can be presumed that the inner side will become thinner because the chamber wall is formed and material transportation is more restricted in the inner side.

**4.2 Pore Formation**

Fine-scale observations from the present study allowed us to reconstruct the actual steps in pore formation. As shown already in previous studies (Bé *et al.*, 1979; Spero, 1988), the structure known as 'pore' in foraminifera is actually a composite structure formed by two opposing wells converging at the POS, one opening towards the outer side located on the OOL and one opening towards the cytoplasm side located on the POS (and same on the IOL). The POS/IOL well has been called the pore plate in previous studies (e.g., Haynes, 1981). These pore plates can also be seen on the organic layer template when fossil foraminiferal tests are dissolved (Bannar *et al*., 1973; Banner and Williams, 1973; Hottinger and Dreher, 1974; Cadere *et al*., 2003; NíNi Fhlaithearta *et al*., 2013). Pore plates seems to be dented at the IOL side, from a previous study (see Fig. 3C in Nagai et al. 2018). Therefore, the pores are not actually pass-through structures formed at once but are instead formed in unison by separate processes on the OOL and the IOL. Pores have been suggested to be used for respiration (e.g., Berthold, 1976; Leutenegger & Hansen, 1979). As $O_2$ and $CO_2$ used in respiration are nonpolar molecules, they are able to pass through the cell membrane. As such, the existence of pore plates made from cytoplasm seen in the present study should not influence respiration. Our observations show that in the initial stage of chamber formation, the pore plate (visible as frustoconical structures of about 1 μm) is already present when the POS is woven, at the growth front. Pore funnels, about 0.5 μm in size, that pair up with the pore plate in the same location (but open to the opposite direction) are formed on the OOL. This structure and the pore plate collectively form the pore, and there is no space between the two for calcium

carbonate to precipitate, and therefore the pore is not calcified. Pore plates and pore funnels smoothly peeled off from one another (Fig. 3B and 7A), suggesting that pore plates and pore funnels belong to independent organic sheets formed from separated pseudopodia. Pore plates and funnels were gently adhered to each other before calcification started. It is thought that pore plates and pore funnels are formed simultaneously face to face, during the organic sheet formation. Pore plates and pore funnels likely function as anchors that hold together all three sheets to result ultimately in a smooth calcareous wall.~~Pore plates and pore funnels are smoothly peeled off each other (Fig. 3B and 7A). It is suggesting that pore plates and pore funnels are belonging to independent organic sheets formed from separated pseudopodia. Pore plates and funnels are gently stuck each other before calcification started. It is thought that pore plate and pore funnel are formed face to face during the organic sheet forming and it would be functioning as anchoring the all three sheet at numerous points of the newly forming chamber to arrange the calcareous wall smooth.~~ 
[revised manuscript text omitted]